# Dr.LLM: Dynamic Layer Routing in LLMs

**Ahmed Heakl**[1,2]*, **Martin Gubri**[1], **Salman Khan**[2], **Sangdoo Yun**[3], **Seong Joon Oh**[1,4,5]*
[1]Parameter Lab, [2]MBZUAI, [3]NAVER AI Lab, [4]University of Tübingen, [5]Tübingen AI Center
`https://github.com/parameterlab/dr-llm`

## Abstract

Large Language Models (LLMs) process every token through all layers of a transformer stack, causing wasted computation on simple queries and insufficient flexibility for harder ones that need deeper reasoning. Adaptive-depth methods can improve efficiency, but prior approaches rely on costly inference-time search, architectural changes, or large-scale retraining, and in practice often degrade accuracy despite efficiency gains. We introduce **Dr.LLM**, Dynamic routing of Layers for LLMs, a retrofittable framework that equips pretrained models with lightweight per-layer routers deciding to *skip*, *execute*, or *repeat* a block. Routers are trained with explicit supervision: using Monte Carlo Tree Search (MCTS), we derive high-quality layer configurations that preserve or improve accuracy under a compute budget. Our design, windowed pooling for stable routing, focal loss with class balancing, and bottleneck MLP routers, ensures robustness under class imbalance and long sequences. On ARC (logic) and DART (math), Dr.LLM improves accuracy by up to +3.4%p while saving 5 layers per example on average. Routers generalize to out-of-domain tasks (MMLU, GSM8k, AIME, TruthfulQA, SQuADv2, GPQA, PIQA, AGIEval) with only 0.85% accuracy drop while retaining efficiency, and outperform prior routing methods by up to +7.7%p. Overall, Dr.LLM shows that explicitly supervised routers retrofit frozen LLMs for budget-aware, accuracy-driven inference without altering base weights.

## 1 Introduction

Large language models (LLMs) typically process every token through a fixed stack of transformer layers, regardless of the input's difficulty. This *static-depth* regime results in wasted computation for easy prompts and insufficient flexibility for challenging reasoning tasks. To address this, prior work has investigated adaptive depth mechanisms at test time, including early-exit strategies (Elhoushi et al., 2024), layer pruning (Men et al., 2024), recurrent or looped blocks (Bae et al., 2025), dynamic routing methods (He et al., 2024; Luo et al., 2025), mixture-of-depth approaches (Raposo et al., 2024), mixture-of-experts architectures (Shazeer et al., 2017), and search-based routing frameworks (Li et al., 2025). Despite their promise, these methods typically suffer from one or more limitations: (i) they trade accuracy for speed, (ii) they require architectural modifications and retraining on substantial amounts of data, or (iii) they rely on costly inference-time search that is difficult to deploy at scale.

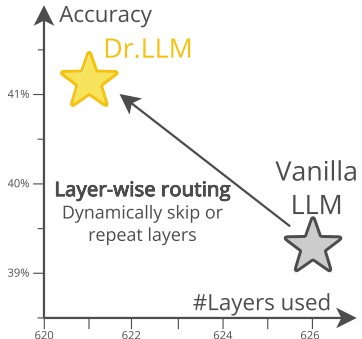

Figure 1: **Dr.LLM improves accuracy while reducing computation.** Number of layers used per example vs. accuracy on ARC and DART, averaged on six models.

We propose **Dr.LLM** (**D**ynamic **R**outing of **L**ayers for **LM**s), a retrofittable framework that equips a frozen, pretrained LLM with lightweight, per-layer routers that decide whether to *skip*, *execute*, or *repeat* their layer. Routers operate in windowed, mean-pooled hidden states and are trained with *explicit supervision* obtained from an offline Monte Carlo Tree Search (MCTS). For each input, MCTS discovers execution paths, that is, which layers to skip or repeat, such that they preserve or improve task accuracy under a constrained compute budget.

---

*Corresponding authors

Training the routers with supervised learning on 4k optimized paths is lightweight, since only the few router parameters are updated while the LLM remains frozen. The trained router removes the need for any search at inference time, enabling compute-efficient inference that increases accuracy, without modifying the base model weights.

Empirically, Dr.LLM improves the accuracy of reasoning-heavy tasks while *reducing* the average number of executed layers (Fig. 1), where "number of layers" denotes the *sum of active layers across all forward passes*, so a 32-layer model generating long sequences accumulates several hundred layer applications in total. Also, Dr.LLM cuts generation time by 15.3% on 1k-token runs, with router overhead under 1%. On ARC (logic) and DART (math), accuracy improves in all cases, with mean gains of **+2.25** percentage points (**%p**) and 5.0 fewer layers per example across six models. Routers generalize out-of-domain (e.g., MMLU, GSM8k, AIME, TruthfulQA, SQuADv2, GPQA, AGIEval, PIQA) with only 0.85%p average accuracy drop while retaining efficiency, indicating that learned routing policies transfer beyond the supervised domains. Lastly, Dr.LLM outperforms all prior SoTA routing methods by up to **+7.7%p** accuracy. Our contributions are as follows:

- **Supervised dynamic routing for frozen LLMs.** We introduce per-layer routers that decide to either `skip`, `execute`, or `repeat` their layer. We train the routers end-to-end on only 4k execution paths optimized for accuracy, discovered offline.

- **Effective path supervision via MCTS.** We present a length-aware MCTS to find layer edits (skips/repeats) under a budget and to retain only accuracy-preserving or improving paths, generating a compact supervision dataset without modifying the base weights.

- **Lightweight router training.** We propose windowed mean-pooling for stable decisions on long contexts and use focal loss with class-rebalancing weights, combined with a lightweight two-layer linear model, which together handle class imbalance and keep the computation overhead negligible.

- **Accuracy increase and compute efficiency.** On ARC and DART across six models, accuracy improves in all cases, with up to **+4.0**%p and **11.0** layers saved per example in that case, without architectural changes, retraining, or inference-time search.

- **Robust generalization.** Routers transfer to out-of-domain benchmarks with only a 0.85%p average drop, showing that policies learned during training remain useful beyond the training tasks.

## 2 RELATED WORK

Adaptive-depth methods span pruning, early exits, recurrence, and routing; Table 1 condenses their trade-offs across accuracy, retrofitting, efficiency, practicality, and frozen-base compatibility.

**Pruning and Early Exit.** Classical model compression prunes redundant weights, heads, or layers post hoc (Sajjad et al., 2023). Early-exit networks extend this by attaching auxiliary classifiers at intermediate layers (Xin et al., 2020; Zhou et al., 2020; Teerapittayanon et al., 2016), letting easy inputs terminate early. While effective, such classifiers need calibration, add overhead, and complicate deployment. LayerSkip (Elhoushi et al., 2024) improves this by training with dropout and a shared exit loss, removing the need for multiple classifiers. Yet it still requires finetuning or training from scratch for the LLM and cannot repeat layers. In contrast, our approach supervises *skip/execute/repeat* directly via MCTS, eliminating auxiliary exits and enabling repetition without retraining base weights.

Table 1: **Comparison of dynamic routing methods for frozen LLMs.** *Accuracy* ↑: does the method improve accuracy over the baseline. *Retrofit*: can it be added to pretrained models with minimal effort. *Cheap I*: enables efficient inference without heavy overhead. *Cheap T*: enables efficient training with limited data. *LLM* ❄: base model remains unchanged. Symbols: ✓= strong support, ✗= not supported. Dr.LLM is the only method satisfying all five criteria.

| Method | Accuracy ↑ | Retrofit | Cheap I | Cheap T | LLM ❄ |
|---|---|---|---|---|---|
| CoLa | ✓ | ✓ | ✗ | ✗ | ✓ |
| Mixture of Depths | ✗ | ✗ | ✓ | ✗ | ✗ |
| Universal Transformer | ✓ | ✗ | ✗ | ✗ | ✗ |
| LLM-Pruner | ✗ | ✓ | ✓ | ✗ | ✗ |
| Mixture of Experts | ✓ | ✗ | ✓ | ✗ | ✗ |
| Mixture of Recursions | ✓ | ✗ | ✗ | ✗ | ✗ |
| LayerSkip | ✗ | ✓ | ✓ | ✗ | ✓ |
| ShortGPT | ✗ | ✗ | ✓ | ✗ | ✗ |
| MindSkip | ✗ | ✓ | ✓ | ✗ | ✓ |
| FlexiDepth | ✗ | ✓ | ✗ | ✗ | ✓ |
| **Dr.LLM (Ours)** | ✓ | ✓ | ✓ | ✓ | ✓ |

Figure 2: **Our layer routing based on hidden states.** Dr.LLM augments a frozen decoder-only LLM with per-layer routers that decide to `skip`, `execute`, or `repeat` a block once. Routers read windowed summaries of hidden states and are trained from MCTS-derived targets (Sec. 4). For clarity, the diagram also highlights the router internals and the flow of hidden states across layers.

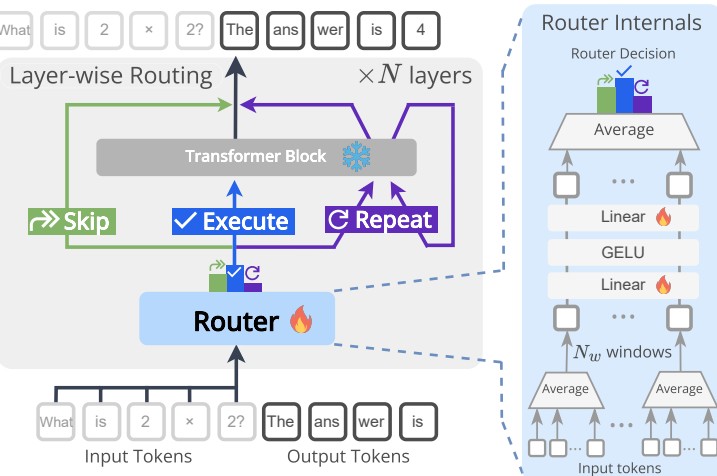

**Recurrence and Looped Architectures.** Another line of work adapts depth by repeating computation. Universal Transformers (Dehghani et al., 2018) learn a halting policy per token, while looped transformers (Yang et al., 2023; Giannou et al., 2023; Geiping et al., 2025) iteratively reapply blocks for refinement ("slow thinking"). These models are flexible but require architectural redesign, full retraining, and incur higher inference cost. We also support targeted repetition, by attaching shallow controllers to frozen layers, avoiding structural changes or pretraining. Moreover, we allow skips to offset the layer increases from looping.

**Dynamic Routing and Modular Inference.** Routing-based methods let inputs select modules dynamically. MoE architectures (Fedus et al., 2022; Shazeer et al., 2017) expand capacity by routing tokens to experts, but demand large-scale retraining. CoLa (Li et al., 2025) is closer to our setting: it treats pretrained layers as modules and searches, via MCTS, for input-specific "chains of layers." However, CoLa requires costly search at inference and, critically, access to gold labels during search to decide which path is "correct," making it impractical for deployment. We instead perform MCTS offline to generate supervision and then train routers that make decisions cheaply at inference. Other adaptive-depth methods, such as FlexiDepth (Luo et al., 2025) and MindSkip (He et al., 2024), retrofit routing to pretrained models but require extensive training (hundreds of thousands of examples) and often reduce accuracy to save compute. By contrast, our routers are trained from only 4k MCTS-derived examples and in experiments improve accuracy while lowering cost. Mixture-of-Depth (MoD) (Raposo et al., 2024) takes a different angle, routing at the *token level* by sending only a subset of tokens through deeper layers, but modifies the base weights. This intra-layer mechanism complements our sequence-level skip/execute/repeat routing: token-level signals identify local redundancy, while layer-level control reallocates global compute.

## 3 SUPERVISED TRAINING OF THE ROUTER

Let a pretrained decoder-only LLM with $L$ transformer blocks be $\mathcal{M} = [\mathcal{B}_1, \ldots, \mathcal{B}_L]$. For a token sequence of length $T$, let $H^{(1)} \in \mathbb{R}^{T \times d}$ denote its initial hidden states. The classical forward pass applies each block once: $H^{(\ell)} = \mathcal{B}_\ell(H^{(\ell-1)})$. We instead seek a discrete per-layer policy

$$y_\ell \in \{\texttt{skip}, \texttt{execute}, \texttt{repeat}\},$$

where `skip` bypasses $\mathcal{B}_\ell$, `execute` applies it once, and `repeat` applies it twice in succession. The vector $\mathbf{y} = (y_1, \ldots, y_L)$ induces a custom execution path, while we freeze the base weights.

### 3.1 ROUTER ARCHITECTURE

As shown in Figure 2, each block $\mathcal{B}_\ell$ is paired with a lightweight MLP (Linear-GELU-Linear) $r_\ell : \mathbb{R}^d \to \mathbb{R}^3$, which outputs logits for $\{\texttt{skip}, \texttt{execute}, \texttt{repeat}\}$. The router operates on a compact summary of the hidden states $H^{(\ell-1)}$ from the previous layer. Routers are executed *once per input sequence at inference*, adding negligible overhead (constant with the number of generated tokens), and remaining fully compatible with KV caching, unlike most layer routing methods.

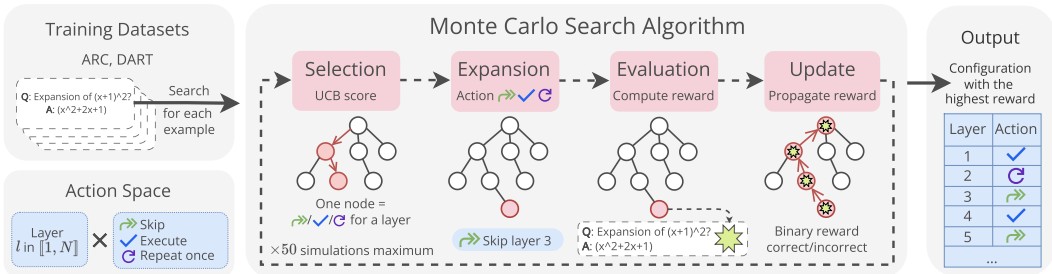

Figure 3: **Length-aware MCTS** used to collect the supervised training dataset of per-layer routing configurations (`skip`/`execute`/`repeat`). For each input, MCTS explores modified layer paths and retains accuracy-preserving or improving ones under a compute budget.

To stabilize decisions on long contexts while keeping overhead negligible, we adopt *windowed mean pooling*: the first $W\lfloor T/W \rfloor$ tokens are divided into contiguous windows $\{S_w\}_{w=1}^W$, with $m_w = \frac{1}{|S_w|} \sum_{t \in S_w} H_t^{(\ell-1)}$ the mean-pooled representation. Router votes are aggregated by averaging logits:

$$z_\ell = \frac{1}{W} \sum_{w=1}^W r_\ell(m_w), \qquad p_\ell = \mathrm{softmax}(z_\ell), \qquad \hat{y}_\ell = \arg \max_{c \in \{0,1,2\}} p_{\ell,c}.$$

We default to $W=8$ (clamped by $T$). Router weights are Xavier-uniform initialized (Glorot & Bengio, 2010) and biases initialized to zero. Only $\{r_\ell\}_{\ell=1}^L$ are trainable, while model parameters are frozen. At inference, the decision $\hat{y}_\ell$ governs block execution where SKIP passes $H^{(\ell)} = H^{(\ell-1)}$, EXECUTE applies $\mathcal{B}_\ell$ once, and REPEAT applies $\mathcal{B}_\ell$ twice in succession.

## 3.2 TRAINING REGIME

For each prompt-response pair / question-answer pair $(q, a)$, the method proposed in Sec. 4 yields a "ground truth" path, $\pi^\star$, that we utilize to supervise the training process introduced here. These paths preserve or improves task reward under a compute budget. We convert $\pi^\star$ to per-layer labels $y_\ell^\star = \mathrm{count}(\ell \in \pi^\star) \in \{0, 1, 2\}$, producing tuples $(q, \mathbf{y}^\star, a)$ meaning (question, $\mathbf{y}^\star$, answer).

Because `execute` dominates, we apply focal loss (Lin et al., 2017) with effective-number weights. Let the global class counts be $n_c$ for $c \in \{\texttt{skip}, \texttt{exec}, \texttt{repeat}\}$ and $\beta \in (0, 1)$:

$$\alpha_c = \frac{1 - \beta}{1 - \beta^{n_c}} \Big/ \frac{1}{3} \sum_{c'} \frac{1 - \beta}{1 - \beta^{n_{c'}}}, \qquad \mathcal{L} = -\frac{1}{L} \sum_{\ell=1}^L \alpha_{y_\ell^\star} (1 - p_{\ell, y_\ell^\star})^\gamma \log p_{\ell, y_\ell^\star}.$$

We use $\gamma=2$, $\beta=0.999$ by default.[1] During training we apply *teacher forcing* for execution only, i.e., we replace the router decision with the ground-truth label $\hat{y}_\ell \leftarrow y_\ell^\star$ to follow the labeled path while supervising logits with $\mathcal{L}$. This avoids making $\mathrm{router}_i$ depend on $\mathrm{router}_{i-1}$ outputs, which would otherwise slow training and lower accuracy by 1.7%. At inference, decisions are greedy: $\hat{y}_\ell = \arg \max p_\ell$; no search is used.

Routers add $O(Ldh)$ parameters for hidden size $d$ and router width $h$ (we use $h=128$), and one small MLP per layer at inference. Windowed pooling is linear in $T$ and inexpensive relative to a transformer block. *Skip* reduces compute; *repeat* adds targeted compute when beneficial. We report accuracy, per-class f1 for $\{\texttt{skip}, \texttt{exec}, \texttt{repeat}\}$, and average executed layers.

## 4 TRAINING DATA GENERATION VIA MCTS

This section describes the search-based generation of the supervised training dataset of layer configurations for the router. We supervise routers using tuples $(q, \mathbf{y}^\star, a)$, where $q$ is the input question/prompt, $a$ the gold answer, and $\mathbf{y}^\star \in \{skip, execute, repeat\}^L$ the *best* per-layer routing targets found by a discrete search over modified forward passes (Fig. 3). The search is offline and does not modify base weights.

---

[1] Setting $\gamma=0$ recovers weighted cross-entropy.

---

**Algorithm 1** Length-aware MCTS for a question–answer pair $(q, a)$

---

**Require:** Default path $\pi_0 = [1, \ldots, L]$, number of simulations $N_s$, constants $(c, \lambda, p_{\text{rand}})$
1: Create root $r$ with $\pi(r) \leftarrow \pi_0$; cache $\mathcal{E} \leftarrow \varnothing$; best path $\pi^\star \leftarrow \varnothing$
2: **for** $n = 1$ to $N_s$ **do**
3:     **Select**: traverse from $r$ to a leaf using nodes UCB score; w.p. $p_{\text{rand}}$ pick a random child
4:     **Expand**: add one untried action to obtain child $u$ (respecting path-length cap)
5:     **Evaluate**: if $\pi(u) \notin \mathcal{E}$, run model constrained to $\pi(u)$, set $\mathcal{E}[\pi(u)] \leftarrow R(\hat{a}, a)$
6:     **Backpropagate**: propagate $R$ to ancestors
7:     **if** $R(\hat{a}, a) = 1$ **and** $|\pi(u)| < |\pi^\star|$ or $\pi^\star = \varnothing$ **then**
8:         $\pi^\star \leftarrow \pi(u)$                                   ▷ update to the shortest correct path
9:         **break if** $\mathcal{E}[\pi_0] = 0$                          ▷ enhance default path answer W→C
10:     **end if**
11: **end for**
12: Convert $\pi^\star$ into per-layer labels $\mathbf{y}^\star \in \{0, 1, 2\}^L$ (skip/execute/repeat)
13: **return** $\mathbf{y}^\star$

---

## 4.1 EDITED EXECUTION PATHS AND ACTIONS

Let the base model have $L$ blocks and default path $\pi_0 = [1, \ldots, L]$. An edited path $\pi = [\ell_1, \ldots, \ell_K]$ preserves the original order of blocks but may omit certain layers (*skip*) or apply a given layer twice (*repeat once*). We allow skips of at most two consecutive layers, and we allow at most a single repeat for any block, which controls the compute growth, i.e., the total edited path length is capped at $|\pi| \leq 2L$.

## 4.2 LENGTH-AWARE MCTS

Each node stores a triple of path $\pi$, visits $v(\pi)$, and cumulative reward $Q(\pi)$. During selection, we maximize a UCB (Upper Confidence Bound) score (inspired by Li et al. (2025)) with an explicit *length penalty* to favor compact paths:

$$\text{UCB}(\pi) = \underbrace{\frac{Q(\pi)}{v(\pi)}}_{\text{exploitation}} + \underbrace{c\sqrt{\frac{\ln V}{v(\pi)}}}_{\text{exploration}} - \underbrace{\lambda \frac{|\pi(\pi)|}{L}}_{\text{length penalty}},$$

where $V$ is the parent's visit count. We use $c=1.8$, $\lambda=3.0$, and with probability $p_{\text{rand}}=0.1$ pick a random child to encourage exploration. For each simulation, we expand one untried action, *evaluate* the edited path once, and backpropagate the task reward $R \in [0, 1]$ (no length penalty) through the root node. We run a fixed budget $N_s=50$ simulations or stop early if we (i) reach correctness and (ii) find a strictly shorter correct path than the best-so-far. Path evaluations are memorized to avoid duplicates. We retain only accuracy-preserving/improving paths (vs. the default $\pi_0$).

We ran MCTS across ARC and DART, collecting 4k supervision examples. About 30% of these edited paths achieved higher accuracy than the default path $\pi_0$, while the rest preserved accuracy and reduce the number

Table 2: **Data generation statistics.** Visited is the total number of candidate paths explored, and Sampled is the subset of paths that improve or preserve accuracy.

| Dataset | Original | Sampled | Visited | #Inferences |
|---------|----------|---------|---------|-------------|
| ARC-E | 2.25k | 400 | 2,090 | 82.6k |
| ARC-C | 1.12k | 600 | 1,119 | 44.2k |
| DART-1 | 117k | 200 | 967 | 38.2k |
| DART-2 | 296k | 400 | 2,242 | 88.6k |
| DART-3 | 364k | 600 | 3,695 | 146.0k |
| DART-4 | 391k | 800 | 6,014 | 237.6k |
| DART-5 | 445k | 1000 | 8,203 | 324.0k |
| **Total** | **1.63M** | **4000** | **24,330** | **961.0k** |

of layers with exact statistic shown in Tab. 2. The average number of layers saved is 1.82. Although the search required 961.0k forward passes, it is performed entirely offline; at inference time, routing decisions are made directly by the trained routers without any search.

Compared to Li et al. (2025), we found that reducing the repetition block size from 4 to 1 made the search substantially faster, while achieving the same accuracy gains and layer savings with only 50 simulations instead of 200. We also found that lowering the length penalty from 5 to 3 reduced the number of search samples by 14.8%p.

## 5 EXPERIMENTS

We evaluate Dr.LLM across both in-domain reasoning tasks and a diverse suite of out-of-domain (OOD) benchmarks to test generalization under distribution shift. Our experimental setup is designed to answer three key questions: (i) Does supervised dynamic routing improve accuracy relative to static baselines? (ii) How much computational efficiency is gained in terms of average executed layers? (iii) Are the learned routing policies robust to new tasks and model families?

**Models.** We retrofit Dr.LLM onto six backbone models spanning two families: LLaMA-3.2 (Dubey et al., 2024) (3B Instruct, 3B Base, 8B Instruct, 8B Base) and Qwen-2.5 (Yang et al., 2024) (3B Instruct, 7B Instruct). These models cover a variety of sizes and both instruction-tuned and base variants.

**Training Data.** Routers are supervised using 4K MCTS-derived tuples (Sec. 4) from ARC-Easy/Challenge (Clark et al., 2018) and DART-Math (Tong et al., 2024). We selected these datasets for three reasons: (1) they provide stratified difficulty levels (ARC-Easy vs. ARC-Challenge, DART-1 to DART-5), (2) they target logic and multi-step mathematical reasoning, where adaptive computation is beneficial, and (3) they have train/test splits allowing us to test in-domain distributions.

**Training Setup.** We train all routers on a single NVIDIA A100 40GB GPU. Given the small number of trainable parameters (11M for 3B models, 0.14% of base weights; 16.8M for 8B models, 0.56%), training is efficient and completes within 4 hours while using only 20% of the GPU VRAM. We use AdamW (Loshchilov & Hutter, 2017) with a cosine schedule, learning rate $1 \times 10^{-3}$, weight decay 0.01, 500 warmup steps, and a total of 25 epochs. The effective batch size is 16, and training is performed in bf16 precision. Routers achieve a macro F1-score of 61%, reflecting class balance across the highly imbalanced actions (skip: 7.6%, repeat: 2.3%), while the actual per-layer accuracy reaches 96.8%. Most errors conservatively predict execute, slightly increasing compute but not harming accuracy, which explains why Dr.LLM consistently improves task performance despite the moderate F1-score. We tested different initialization schemes for router biases, (i) empirical class frequencies and (ii) zero-initialization, finding the latter more stable and yielding stronger downstream accuracy.

**Wall-clock latency and router overhead** We measure end-to-end generation time for sequences of 1,000 tokens. Dr.LLM achieves a **15.3%** wall-time reduction (29.21 s vs. 34.49 s baseline), while the router adds only **0.27** s per query (<1% of total latency).

**In-Domain Evaluation.** We first evaluate routers on ARC and DART test splits. These tasks serve as a direct measure of whether routers can recover the MCTS supervision signal and yield improvements under controlled conditions.

**Out-of-Domain Evaluation.** To assess robustness, we evaluate the router-equipped models on a broad range of benchmarks: MMLU (Hendrycks et al., 2020) for factual knowledge, GSM8k (strict_match) (Cobbe et al., 2021) for grade-school math, TruthfulQA (mcq1) (Lin et al., 2021) for adversarial factuality, GPQA Diamond (Rein et al., 2024) and AIME24 (MAA, 2024) for challenging mathematical reasoning, AGIEval (Zhong et al., 2023) for exam-style reasoning, SQuADv2 (f1) (Rajpurkar et al., 2018) for reading comprehension, and PIQA (Bisk et al., 2020) for commonsense reasoning. All benchmarks are reported using acc_norm computed from log-likelihoods in the lm-eval-harness framework (Gao et al., 2024), except GSM8k, TruthfulQA, and SQuADv2 which follow their respective metrics. Evaluations are run with default settings, maximum generation length of 2048 tokens, and greedy decoding.

## 6 RESULTS & DISCUSSION

We evaluate Dr.LLM on in-domain tasks, test its robustness on out-of-domain benchmarks, and analyze routing patterns with ablations.

### 6.1 IN-DOMAIN PERFORMANCE ON ARC AND DART

Table 3 summarizes in-domain results on ARC (logic) and DART (math), showing that routers consistently improve accuracy while reducing the average number of layers executed across all six

Table 3: **Routers consistently improve accuracy and reduce executed layers across all models.** In-domain results on ARC (logic) and DART (math). Accuracy in %. (+x) indicates accuracy gains, (-x) indicates layer savings, and (+0.0) indicates no change.

| Model | ARC | | DART | | Total | |
|---|---|---|---|---|---|---|
| | Accuracy | Num Layers | Accuracy | Num Layers | Accuracy | Num Layers |
| LLaMA-3B-Instruct | 73.5 | 103.9 | 35.2 | 422.0 | 46.1 | 331.1 |
| + Router | **74.5** (+1.0) | 99.50 (-4.25) | **38.6** (+3.4) | 413.3 (-8.66) | **48.9** (+2.7) | 323.7 (-7.40) |
| LLaMA-8B-Instruct | 88.5 | 106.9 | 38.4 | 320.0 | 52.7 | 518.2 |
| + Router | **89.4** (+0.9) | 104.0 (-2.94) | **41.2** (+2.8) | 309.1 (-10.96) | **54.7** (+2.3) | 509.6 (-8.66) |
| LLaMA-3B-Base | 48.0 | 56.0 | 11.8 | 548.4 | 22.1 | 815.4 |
| + Router | **49.0** (+1.0) | 55.70 (-0.28) | **15.8** (+4.0) | 544.3 (-4.12) | **25.3** (+3.2) | 812.4 (-3.02) |
| LLaMA-8B-Base | 22.5 | 56.0 | 17.2 | 536.7 | 18.7 | 798.7 |
| + Router | **23.5** (+1.0) | 55.60 (-0.42) | **20.2** (+3.0) | 531.0 (-5.74) | **21.1** (+2.4) | 794.5 (-4.22) |
| Qwen-3B-Instruct | 53.0 | 115.7 | 30.2 | 536.3 | 36.7 | 832.4 |
| + Router | **55.5** (+2.5) | 115.5 (-0.23) | **32.4** (+2.2) | 531.7 (-4.55) | **39.0** (+2.3) | 828.9 (-3.31) |
| Qwen-7B-Instruct | 94.5 | 112.0 | 45.4 | 277.8 | 59.4 | 460.9 |
| + Router | 94.5 (+0.0) | 111.8 (-0.20) | **46.8** (+1.4) | 273.1 (-4.67) | **60.4** (+0.9) | 457.5 (-3.39) |

Table 4: **Generalization to out-of-domain benchmarks**. Accuracy in %. Router models maintain accuracy with 0.85%p average drop while preserving efficiency. All evaluated models are instruct, we use LLaMa-3.2 and Qwen2.5. TQA is TruthfulQA, and GPQA D is GPQA Diamond.

| Model | MMLU | AIME24 | TQA | GSM8k | SQuADv2 | GPQA D | AGIEval | PIQA | Avg. $\Delta$ |
|---|---|---|---|---|---|---|---|---|---|
| LLaMA3B | $60.5_{\pm0.39}$ | $3.3_{\pm1.33}$ | $31.3_{\pm1.48}$ | $64.9_{\pm1.32}$ | $32.6_{\pm1.41}$ | $27.2_{\pm0.31}$ | $35.7_{\pm0.51}$ | $75.6_{\pm1.06}$ | - |
| + Router | $59.5_{\pm0.40}$ | $3.3_{\pm1.64}$ | $30.4_{\pm1.31}$ | $64.3_{\pm1.35}$ | $30.6_{\pm1.42}$ | $29.8_{\pm0.33}$ | $33.8_{\pm0.50}$ | $71.9_{\pm1.07}$ | -0.94 |
| LLaMA8B | $67.9_{\pm0.72}$ | $6.7_{\pm1.75}$ | $36.9_{\pm1.45}$ | $73.2_{\pm1.30}$ | $29.1_{\pm0.35}$ | $34.3_{\pm0.31}$ | $43.2_{\pm0.52}$ | $80.9_{\pm1.06}$ | - |
| + Router | $66.8_{\pm0.70}$ | $6.7_{\pm1.74}$ | $36.6_{\pm1.40}$ | $74.9_{\pm1.28}$ | $28.6_{\pm0.35}$ | $32.3_{\pm0.41}$ | $41.5_{\pm0.51}$ | $79.2_{\pm1.07}$ | -0.70 |
| Qwen3B | $65.3_{\pm0.82}$ | $6.7_{\pm1.36}$ | $41.9_{\pm1.50}$ | $11.1_{\pm1.29}$ | $21.5_{\pm0.99}$ | $33.3_{\pm0.34}$ | $54.2_{\pm0.51}$ | $78.1_{\pm1.05}$ | - |
| + Router | $62.8_{\pm0.82}$ | $6.7_{\pm1.38}$ | $41.9_{\pm1.47}$ | $11.5_{\pm1.29}$ | $20.1_{\pm0.81}$ | $32.4_{\pm0.35}$ | $49.4_{\pm0.51}$ | $78.9_{\pm1.04}$ | -1.05 |
| Qwen7B | $71.7_{\pm0.88}$ | $10.0_{\pm1.40}$ | $47.7_{\pm1.53}$ | $75.6_{\pm1.26}$ | $20.8_{\pm0.42}$ | $32.8_{\pm0.36}$ | $61.2_{\pm0.51}$ | $79.7_{\pm0.93}$ | - |
| + Router | $71.2_{\pm0.88}$ | $10.0_{\pm1.42}$ | $47.9_{\pm1.55}$ | $75.7_{\pm1.25}$ | $20.2_{\pm0.43}$ | $32.8_{\pm0.32}$ | $57.2_{\pm0.52}$ | $78.8_{\pm0.92}$ | -0.70 |

models. On ARC, gains are modest (+0.9–2.5%p), reflecting that logic questions already require relatively shallow reasoning. In contrast, DART exhibits larger improvements (+1.4–4.0%p), where the router often assigns `repeat` to late layers, effectively allocating more computation to iterative refinement needed for multi-step math problems. For example, LLaMA-3B-Base improves from 11.8% to 15.8% accuracy (+4.0%p) while saving 4.12 layers per query on average, and Qwen-3B-Instruct gains +2.2%p while cutting 4.6 layers per query. Notably, instruction-tuned models start with substantially higher accuracy than their base counterparts, yet still benefit from routing: e.g., LLaMA-8B-Instruct improves by +2.8%p on DART while saving 11.0 layers per query on average. Importantly, Dr.LLM never degrades accuracy and always saves inference compute with 3–11 fewer layers per query. These results demonstrate that Dr.LLM not only reduces computation but also improves accuracy, with the largest benefits on tasks requiring deeper or repeated reasoning steps.

## 6.2 GENERALIZATION TO OUT-OF-DOMAIN BENCHMARKS

Table 4 evaluates Dr.LLM-equipped models on a diverse suite of out-of-distribution benchmarks, from in-domain and out-of-distribution mathematical reasoning benchmarks (AIME24, GSM8k) to out-of-domains benchmarks specialised in knowledge (MMLU, AGIEval, GPQA Diamond), factuality (TruthfulQA), comprehension (SQuADv2), and commonsense (PIQA). Despite not trained to handle these types of questions, the routers maintain a good generalization with 0.85%p average accuracy drop across the eight benchmarks and four instruct models. The routers decision generalizes to other in-domain benchmarks: all four models gain an average 0.40%p accuracy on GSM8k and maintain the exact same accuracy on AIME24, while reducing compute. In out-of-domain benchmarks, the accuracy drop is limited to 1.20%p on average. Notably, in some cases routers even improve accuracy, such as GPQA Diamond with LLaMA-3B (+2.5%p). In all cases, the router main-

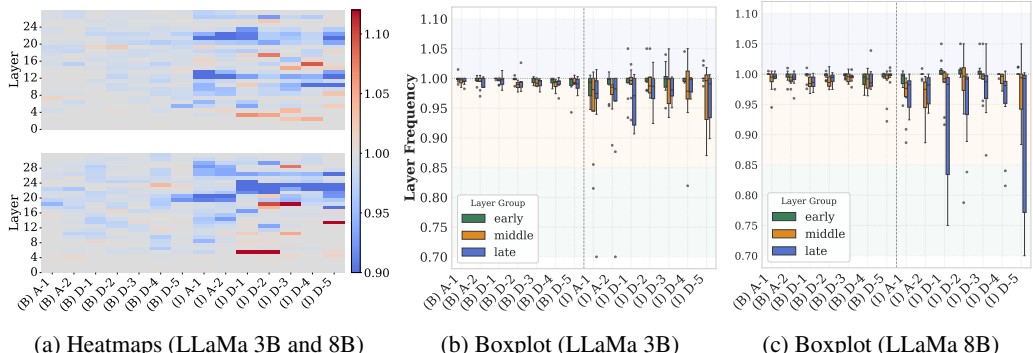

(a) Heatmaps (LLaMa 3B and 8B)  (b) Boxplot (LLaMa 3B)  (c) Boxplot (LLaMa 8B)

Figure 4: **Analysis of routing decisions per layer, dataset, and model.** *(a)* Layer frequency of LLaMa 3B and 8B base (*B*) and instruct (*I*) models across ARC and DART. *(b,c)* Layer frequency grouped by early, middle, and late layers. The x-axis corresponds to the dataset difficulty levels: ARC-Easy (A-1), ARC-Challenge (A-2), and DART levels 1–5 (from D-1 to D-5).

tains its efficiency by saving layers. These results indicate that router policies transfer beyond their domain, suggesting that the learned skip and repeat patterns capture general structural redundancies in transformer computation. Thus, Dr.LLM not only yields efficiency and accuracy improvements in-domain, but also preserves robustness when deployed to unseen, distribution-shifted benchmarks.

## 6.3 COMPARISON TO EXISTING METHODS

Most adaptive-depth approaches either sacrifice accuracy for efficiency or impose costly architectural changes. For example, in Tab. 5, FlexiDepth (Luo et al., 2025) saves four layers on LLaMA-8B but suffers a −6.1%p accuracy drop on GSM8k, while MindSkip (He et al., 2024) reduces compute yet loses −7.8%p on HumanEval. ShortGPT (Men et al., 2024) also improves efficiency but underperforms on reasoning, reaching only 53.6% on GSM8k compared to Dr.LLM's 74.9%. Even FlexiDepth, the method closest in accuracy to Dr.LLM, requires training: it is

Table 5: **Comparison of Dr.LLM with existing methods on reasoning and coding benchmarks.** Results on LLaMa3-8B reported from FlexiDepth (Luo et al., 2025) with 4 layers saved. Although these benchmarks are *in-domain* for prior methods and *out-of-domain* for Dr.LLM, ours still achieves the highest accuracy.

| Method | GSM8k | MMLU | HellaSwag | HumanEval | Avg. |
|---|---|---|---|---|---|
| LayerSkip | 0.4 | 65.9 | 63.6 | 0.0 | 32.5 |
| ShortGPT | 53.6 | 66.4 | 66.2 | 9.2 | 48.9 |
| MindSkip | 37.8 | 66.4 | 69.8 | 18.9 | 48.2 |
| FlexiDepth | 65.7 | 66.3 | 74.3 | 32.3 | 59.7 |
| Dr.LLM | **74.9** | **66.8** | **79.3** | **48.6** | **67.4** |

trained on Tulu-v2 (Ivison et al., 2023) with 326k examples, incurring substantial compute. By contrast, Dr.LLM achieves higher accuracy with far lower overhead, trained on only 4k MCTS-derived examples using a single GPU, *despite the fact that these benchmarks are in-domain for prior routing methods but out-of-domain for Dr.LLM*.

## 6.4 ANALYSIS OF LAYER ROUTING PATTERNS

We analyze router decisions across layers, models, and datasets to identify which layers can be skipped and which improve accuracy when repeated. Fig. 4 visualizes the learned routing policies for LLaMA 3B and 8B models. The heatmaps (Fig. 4a) show structured patterns rather than random skipping: early layers are consistently executed close to once, middle layers are frequently down-weighted, and late layers are often repeated, especially on reasoning-intensive DART tasks. The boxplots (Figs. 4b,4c) confirm this trend: early layers exhibit the lowest variance in execution frequency (stable usage), middle layers show wider skip distributions, and later layers are biased toward repeated execution, indicating their role in iterative refinement. This effect is stronger in the 8B model, where late-layer repetition dominates, suggesting that larger models rely more heavily on additional depth for complex reasoning. Together, these results indicate that Dr.LLM learns routing behaviors aligned with transformer computation phases: maintaining stability in early input process-

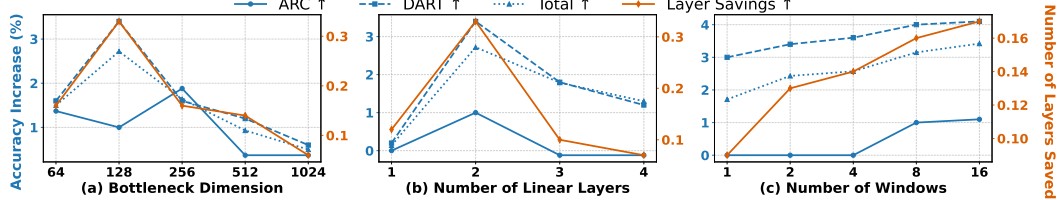

Figure 5: **Ablation study**. We apply Dr.LLM on LLaMa3.2-3B and control: (a) the effect of bottleneck dimension, (b) the effect of number of linear layers, and (c) the effect of number of windows.

ing, economizing in middle layers, and reinvesting compute in later blocks where deeper reasoning is most beneficial.

## 6.5 ABLATION STUDIES

**Router internals.** We ablate the router components to understand their effect on accuracy and efficiency (Fig. 5). Varying the bottleneck dimension (Fig. 5a) shows that smaller hidden sizes (64–128) strike the best balance: a bottleneck of 128 yields the highest accuracy gains (+3.4%p), while larger dimensions reduce both accuracy and layer savings, likely due to overfitting. Next, tuning the number of linear layers (Fig. 5b) indicates that both accuracy and compute gains are best when the router is composed of two linear layers. Deeper routers fail to improve routing, confirming that compact routers are more suitable. The number of pooling windows (Fig. 5c) strongly influences accuracy gains: more windows consistently increase both the accuracy and the number of layers saved. Averaging the hidden states of all input tokens is a signal that is too coarse for the router to learn. Finally, the focal loss better accounts for the class imbalance of the router labels than the weighted cross-entropy loss (+1.1%p. on ARC and +1.8%p. on DART). These trends highlight that Dr.LLM benefits from (i) a compact router architecture, and (ii) windowed contexts to learn fine-grained hidden state features.

**What do the routers learn?** Since the semantics of the question is predictive of LLM accuracy in this question (Ulmer et al., 2024), we ask if the routers truly learn from the internal state of the model or from the types of input. Routers could learn question patterns (e.g. skip the seventh layer for math questions). Table 6 reports the ARC and DART accuracies of a router trained on input embeddings (for all layers), rather than on the hidden states of the previous layer. This new router performs considerably worse than Dr.LLM (-8.6%p on DART), and even worse than the vanilla model without layer routing

Table 6: **Dr.LLM routes layers from their state, not from the type of question.** Benchmark accuracy of routers trained on the hidden states of the previous-layer or of the first layer. In %.

| Router Features | ARC | DART |
|---|---|---|
| Prev. layer $H_{i-1}$ (**Dr.LLM**) | **74.5** | **38.6** |
| First layer $H_1$ (embeddings) | 70.9 | 30.0 |
| No routing (vanilla model) | 73.5 | 35.2 |

(-5.2%p). Therefore, Dr.LLM learns to dynamically map the internal model states to the decision to skip or repeat layers, instead of relying on shallow static signals from the inputs.

## 6.6 BATCH PROCESSING EFFICIENCY

A natural concern with per-layer routing is whether skip/repeat actions disrupt batch processing. We address this by synchronizing per-layer: sequences are grouped by their routing decision (skip/execute/repeat) at each layer and processed together as a *microbatch*. When a repeat decision occurs, all sequences must join the group to complete both passes through the layer before proceeding. However, the synchronization overhead is low because repeat actions occur in only 2.3% of layers. The high routing consensus (80-89% agreement) ensures that most layers process the majority of sequences in a single unified batch, maintaining high GPU utilization.

Table 7 reports throughput measurements (tokens/second) on ARC-E, ARC-C, and DART 1-5 datasets using LLaMA-3B-Instruct. Dr.LLM achieves an average improvement of 8.65% across all batch sizes. Importantly, Dr.LLM always increases throughput, showing that synchronization overhead is not a problem even for large batch sizes.

Table 7: **Dr.LLM consistently improves throughput across all batch sizes.** Throughput measurements (tokens/second) on ARC-E, ARC-C, and DART 1-5 using LLaMA-3B-Instruct. Despite per-layer synchronization requirements, Dr.LLM achieves 8.65% average improvement.

| Batch Size | Original Throughput | Dr.LLM Throughput | Improvement |
|:---:|:---:|:---:|:---:|
| 1 | 25.05 | 31.78 | 26.89% |
| 2 | 37.97 | 41.87 | 10.29% |
| 4 | 70.74 | 76.12 | 7.60% |
| 8 | 107.74 | 115.56 | 7.25% |
| 16 | 117.86 | 122.89 | 4.26% |
| 32 | 125.43 | 130.83 | 4.31% |
| 64 | 103.18 | 108.05 | 4.72% |
| 128 | 109.12 | 113.38 | 3.90% |
| **Average** | 87.14 | **92.56** | **8.65%** |

**Decision Consensus.** The key insight explaining why synchronization overhead remains minimal is that sequences naturally exhibit high routing consensus. We measured inter-prediction similarity across all six models in our study, finding that routing decisions agree 80-89% of the time across different sequences (Table 8). This high similarity means that at most layers, sequences make identical skip/execute/repeat decisions, allowing our microbatching strategy to group them efficiently.

Table 8: **Routing decision consensus across models.** Inter-prediction similarity measures the percentage of identical skip/execute/repeat decisions made by routers across different sequences.

| Model | Similarity (%) |
|:---|:---:|
| LLaMA-3B-Instruct | 80.47 |
| LLaMA-8B-Instruct | 86.20 |
| LLaMA-3B-Base | 80.86 |
| LLaMA-8B-Base | 83.49 |
| Qwen-3B-Instruct | 89.41 |
| Qwen-8B-Instruct | 85.26 |

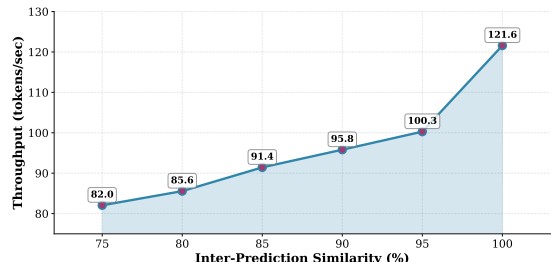

Figure 6: **Batching efficiency increases with routing consensus.** Higher inter-prediction similarity (agreement in skip/execute/repeat decisions across sequences) translates to improved throughput.

This consensus directly drives batching efficiency. Batches with 95% consensus achieve 100.28 tokens/sec, while perfect consensus reaches 121.58 tokens/sec (Table 6), demonstrating that higher routing agreement naturally improves parallel processing. Despite per-layer synchronization, high routing consensus and infrequent repeats enable Dr.LLM to achieve both dynamic depth allocation and parallel efficiency. Computational savings from skipped and optimally routed layers outweigh synchronization overhead, yielding consistent throughput improvements across all batch sizes.

## 7 CONCLUSION

We introduced Dr.LLM, a retrofittable framework that equips frozen LLMs with lightweight routers for *skip/execute/repeat* decisions. Supervised on high-quality paths from length-aware MCTS, Dr.LLM removes inference-time search and architectural changes while improving both efficiency and accuracy. On ARC and DART, it yields up to +3.4%p accuracy with 3–11 layers saved per query, outperforms prior routing methods by up to +7.7%p, and generalizes to out-of-domain benchmarks with only a 0.85%p drop. Routing analysis reveals structured patterns, early layers preserved, middle pruned, late reused, showing that adaptive compute allocation is both learnable and aligned with transformer computation phases. Overall, Dr.LLM demonstrates that explicit supervised routing reconciles efficiency, accuracy, and robustness without retraining, providing a practical step toward budget-aware reasoning and scalable adaptive inference.

ACKNOWLEDGEMENTS

This work was supported by the NAVER corporation.

The authors are also grateful to Cornelius Emde for his careful proofreading.

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

APPENDIX

## A  AUTHOR CONTRIBUTIONS

**All authors** contributed to writing and editing the paper.

**Ahmed Heakl** proposed the initial idea and motivation for the work, drafted the experimental settings, implemented and ran all experiments, collected the data, analyzed results, prepared visualizations, reviewed related work, wrote the first draft, edited the paper, and published the research artefacts.

**Martin Gubri** provided daily supervision, helped refine the experimental settings, consolidated the paper's narrative, proposed the out-of-distribution evaluation (Section 6.2) and the "What do the routers learn?" experiment (Table 6), offered technical support, contributed to the initial draft, and reviewed the final draft.

**Ahmed Heakl** and **Martin Gubri** jointly created the diagrams (Figures 1–3).

**Salman Khan** provided feedback during the ideation phase and reviewed the draft.

**Seong Joon Oh** and **Sangdoo Yun** suggested Figures 1 and 2, validated the experimental settings, and contributed to writing and editing the paper.

**Martin Gubri**, **Seong Joon Oh**, and **Sangdoo Yun** provided weekly supervision and supported the project through organizational and funding contributions.

## B  SCORING, REWARD, AND ANSWER CHECKING

Given an input $q$ and a candidate path $\pi$, we run generation with the model constrained to $\pi$ and obtain a textual response $\hat{a}$. We then map $\hat{a}$ to a scalar reward $R(\hat{a}, a)$:

- **ARC (multi-choice).** Extract a letter $A$–$D$ via a strict regex match (accepting optional "Answer:"). The reward is 1 for a correct letter, 0 otherwise.

- **DART (math).** Extract the boxed expression $\boxed{\cdot}$ (inject if needed for base models), then compute $R = \text{grade\_answer}(\hat{a}, a) \in [0, 1]$ using a symbolic equivalence checker and a robust string comparator.

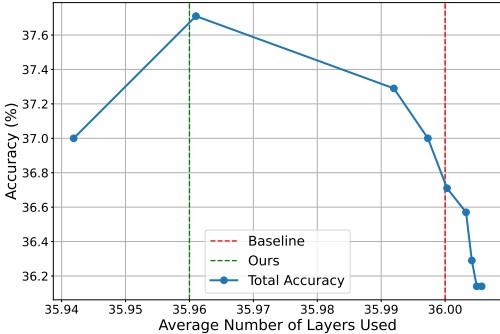

(a) Accuracy vs. average layers used under control interpolation.

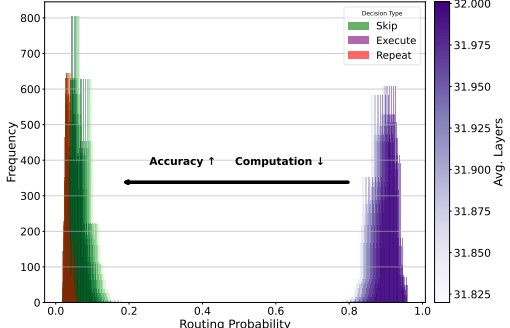

(b) Routing probability distribution across skip/execute/repeat.

Figure 7: **Fine-grained control in LLaMA-8B.** (a) Accuracy as a function of interpolated routing decisions, compared to baseline (red) and ours (green). (b) Histogram of routing probabilities. Shifts from execute $\rightarrow$ skip correlate with higher accuracy, while repeat allocations increase computation.

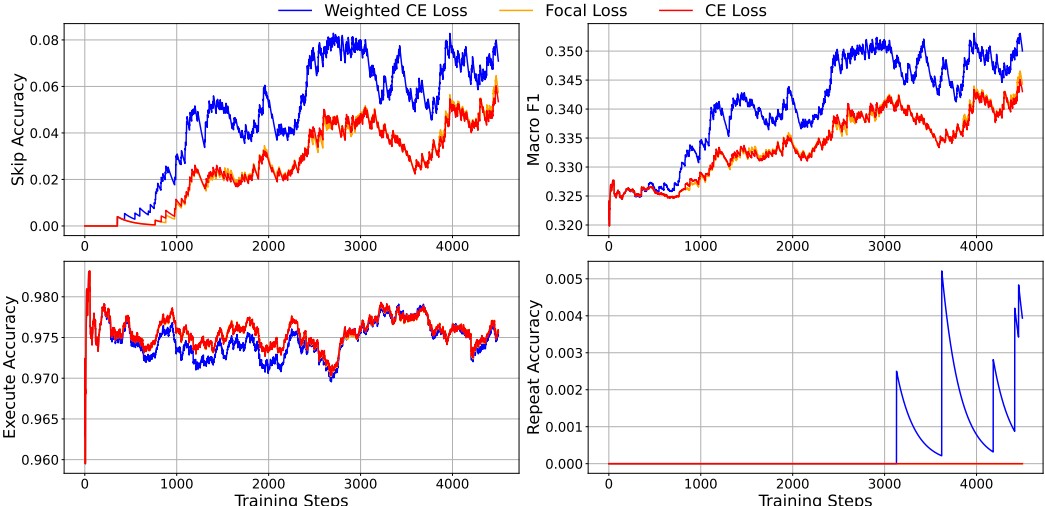

Figure 8: **Effect of loss choice under class imbalance.** Macro F1 across training for weighted CE, focal, and plain CE. While all losses perform similarly on the majority `execute` class, only focal loss improves `skip` accuracy and yields non-trivial `repeat` accuracy, highlighting its necessity for minority classes.

## C    FINE-GRAINED CONTROL OF ROUTER DECISIONS

Beyond analyzing learned routing policies, we study whether router decisions can be *continuously controlled* to balance accuracy and efficiency. Figure 7 reports results for LLaMA-8B.

We introduce a scalar control parameter $p \in [-1, 1]$ that interpolates router probabilities with fixed `skip`, `execute`, or `repeat` distributions:

$$\pi(p) = \begin{cases} (1-t)\,\pi_{\text{skip}} + t\,\pi_{\text{router}}, & p \in [-1, -0.5], \ t = \frac{p+1}{0.5}, \\ (1-t)\,\pi_{\text{router}} + t\,\pi_{\text{exec}}, & p \in (-0.5, 0.5], \ t = \frac{p+0.5}{1.0}, \\ (1-t)\,\pi_{\text{exec}} + t\,\pi_{\text{repeat}}, & p \in (0.5, 1], \ t = \frac{p-0.5}{0.5}. \end{cases}$$

Here $\pi_{\text{router}}$ are the learned router probabilities, and $\pi_{\text{skip}}, \pi_{\text{exec}}, \pi_{\text{repeat}}$ are one-hot distributions over the three actions.

This formulation allows $p$ to smoothly traverse the spectrum from aggressive skipping to repeated execution, without retraining the router. Figure 7a shows that modest interpolation ($p \approx -0.5$) reduces average layers while slightly *increasing* accuracy, suggesting that routers tend to over-execute by default. The distributional shifts in Figure 7b corroborate this: reallocating mass from `execute` toward `skip` correlates with accuracy gains, while reallocating toward `repeat` primarily increases computation with diminishing benefit.

In sum, router behavior is not only learnable but also tunable post-training, enabling fine-grained control over the accuracy–efficiency trade-off through a single scalar knob.

## D    FOCAL VS. CROSS-ENTROPY UNDER CLASS IMBALANCE

Router supervision is highly imbalanced: $n_{\text{skip}} = 4{,}399$, $n_{\text{execute}} = 120{,}956$, $n_{\text{repeat}} = 1{,}457$. Plain cross-entropy minimizes error by predicting the dominant `execute` class, yielding trivial accuracy on `skip`/`repeat`. Weighted CE partly compensates, but still collapses on `repeat`. Focal loss (Lin et al., 2017) reweights classes and down-modulates easy majority examples, forcing learning on rare actions. As shown in Fig. 8, all losses perform similarly on `execute`, but focal substantially improves `skip` accuracy and is the only setup where non-trivial `repeat` accuracy is learned. Thus, focal loss is essential to mitigate imbalance and enable useful skip/repeat routing.

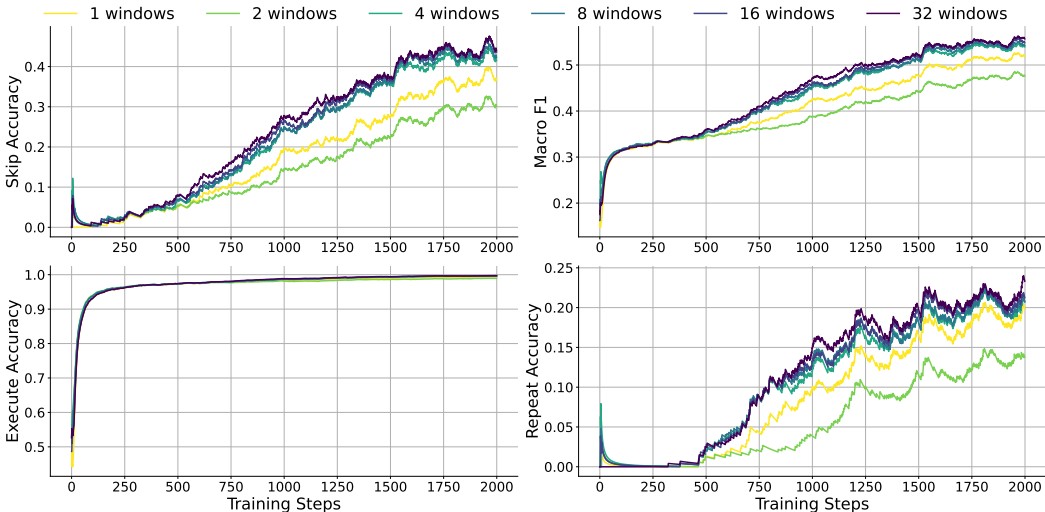

Figure 9: **Effect of window size on router training.** Larger pooling windows consistently improve minority-class accurac. Gains saturate beyond 16 windows, suggesting diminishing returns.

## E    TRAINING ON MORE WINDOWS

Windowed mean pooling stabilizes router decisions by aggregating hidden states over larger contexts. Figure 9 shows that increasing the number of windows yields consistent improvements for minority actions. Skip accuracy rises from $0.32$ (1 window) to $0.42$ (32 windows), and repeat accuracy nearly doubles from $0.12$ to $0.23$. Execute accuracy stays unchanged at $> 0.98$, confirming that the majority class is unaffected. Macro-F1 improves from $0.42$ to $0.53$, with most of the gain realized between 8 and 16 windows, indicating that more granular context summaries significantly help routers capture rare actions without harming the dominant class.

## F    MCTS TRAINING DATA ANALYSIS

**Labels distribution.** To better understand the supervision signal provided to the routers, we analyze the distribution of skip/execute/repeat actions across datasets and model families (Fig. 10). Across all models and datasets, the vast majority of labels are execute, typically exceeding 90%, confirming the extreme class imbalance ($n_{\text{execute}} \gg n_{\text{skip}}, n_{\text{repeat}}$) and motivating focal loss with rebalancing during training (Sec. D). Skip ratios vary across datasets: ARC-Easy and ARC-Challenge exhibit noticeably higher skip counts than DART, suggesting that logical reasoning tasks permit redundancy while mathematical reasoning tasks require more thorough computation. Repeats are rare overall (1–3% of labels) but occur consistently across all datasets, with higher frequency in more challenging DART levels, indicating that repetition is a targeted mechanism for difficult problems rather than a generic operation. Model family and scale also influence distributions: LLaMA-Base models exhibit more balanced skip/execute ratios compared to their instruction-tuned counterparts, which strongly favor execution, while instruction-tuned variants slightly increase repeat counts. Larger 8B models reduce skips further, reflecting greater reliance on their depth, though still allocating some repeats when beneficial. Overall, the MCTS-derived labels capture structured, interpretable routing signals under heavy imbalance, requiring routers to learn policies where most layers execute but the rare skip and repeat actions play a disproportionate role in efficiency and accuracy.

**Decisions per layer.** Figure 11 reveals structured routing patterns that align with transformer computation phases. Across all model families, early layers (embedding and low-level processing) are almost always executed, indicating their necessity for stable representations. Middle layers show the highest variation, with frequent skips reflecting redundancy in feature composition. Late layers display higher repeat frequencies, particularly for the more difficult DART tasks, suggesting

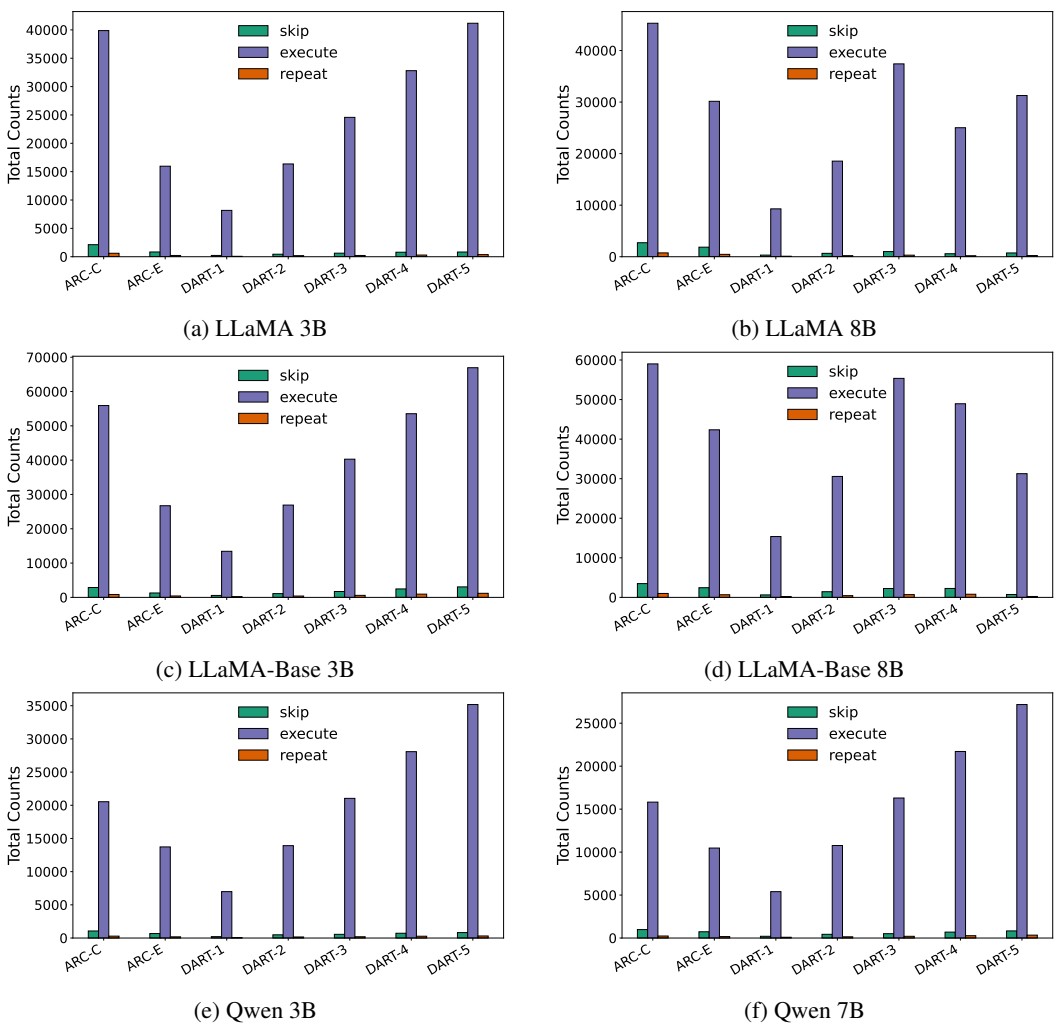

Figure 10: **Label distribution across models.** Distribution of skip/execute/repeat actions across datasets for different planners: (a) LLaMA-3B, (b) LLaMA-8B, (c) LLaMA-Base-3B, (d) LLaMA-Base-8B, (e) Qwen-3B, (f) Qwen-7B.

that deeper refinement is allocated where multi-step reasoning is required. Instruction-tuned models exhibit more aggressive skipping than base models, supporting the view that fine-tuning creates functionally specialized layers that routers can prune more confidently. These trends confirm that Dr.LLM learns consistent, interpretable depth allocation policies across both model scale and family.

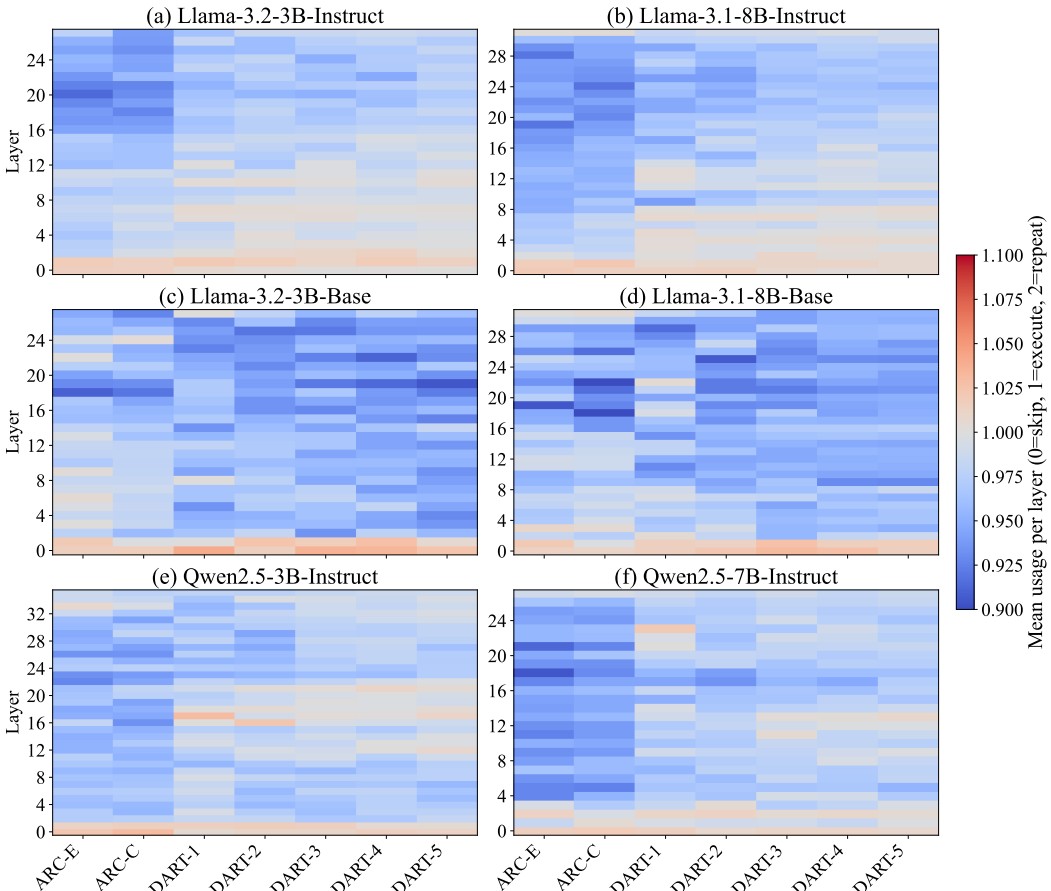

Figure 11: **Per-layer routing frequency across datasets and models.** Heatmaps show the mean usage per layer (0 = skip, 1 = execute, 2 = repeat) for six backbones: (a) LLaMA-3.2-3B-Instruct, (b) LLaMA-3.1-8B-Instruct, (c) LLaMA-3.2-3B-Base, (d) LLaMA-3.1-8B-Base, (e) Qwen2.5-3B-Instruct, and (f) Qwen2.5-7B-Instruct. The x-axis corresponds to benchmark subsets (ARC-E, ARC-C, DART1–5). Early layers are consistently executed, middle layers are frequently skipped, and late layers are occasionally repeated, especially on more complex DART levels.

