# OpenReview forum: "Dr.LLM: Dynamic Layer Routing in LLMs"
_ICLR.cc/2026/Conference — ICLR 2026 Poster_

### Official Review · Reviewer_kHKr · 2025-10-16

**Soundness:** 3
**Presentation:** 3
**Contribution:** 2
**Rating:** 6
**Confidence:** 3

**Summary:**

This paper introduces Dr.LLM, a dynamic layer routing method for large language models (LLMs) that operates at the sequence level. The motivation behind this approach arises from the limitations of traditional static-depth LLMs, which apply a fixed number of layers regardless of input complexity. Such models tend to waste computation on simple prompts and lack the flexibility needed to handle more challenging reasoning tasks. Dr.LLM addresses this by introducing a lightweight routing module at each transformer layer. Based on the sequence of embeddings from the previous layer, the router dynamically decides whether to skip the current layer, execute it once, or execute it twice. These routing decisions are supervised using a Monte Carlo Tree Search (MCTS)-based algorithm. The paper presents experimental results on both in-domain tasks, where the input distribution matches the data used to train the router, and out-of-domain tasks, which involve distribution shifts. Dr.LLM is also evaluated against prior routing methods.

**Strengths:**

* The paper is clearly written, and the authors present their method in a precise and comprehensible way.
* The use of MCTS to generate routing paths is unique and answers the requirements of a routing method presented in the introduction section.
* The method outperforms other routing strategies.

**Weaknesses:**

* The computational cost associated with generating the MCTS-based supervision is not sufficiently reported. While this supervision is computed offline, its upfront cost does not appear to be negligible. It would be valuable to include a comparison of this cost relative to the resources required to train the router itself. Additionally, since performance gains have been demonstrated primarily on in-domain data, expanding the router’s training data to cover a broader range of tasks seems essential for making Dr.LLM applicable to general-purpose LLM scenarios and therefore understanding the computational effort required to generate such supervision is important.
* KV cache compatibility — The authors state at the beginning of Section 3.1 that the method is KV cache compatible, but it's unclear how this is achieved. During generation, the router can select a different path at each forward pass, potentially resulting in different inputs to each transformer block. This variability seems to conflict with standard KV cache usage, which assumes consistent execution paths. To clarify, consider the following example: suppose we are generating a sequence and have reached the point of generating the 100th token. At this step, we pass $H^{\ell-1}$ through the router and obtain $y_{\ell}=\text{skip}$. However, in the next forward pass (i.e., when generating the 101st token), the router outputs $y_{\ell} = \text{exec}$. In this case, how is the KV cache for layer $\ell$ defined to correctly support both scenarios? The mechanism for maintaining cache consistency under dynamic routing decisions needs further clarification.
* The efficiency analysis in the experimental section appears incomplete. Is the overhead introduced by the routing modules truly negligible compared to the reported efficiency gains (0.9–2.5%) in Table 3?

**Questions:**

* A more precise definition of the per-example number-of-layers-used metric would be helpful. I assume this refers to summing the number of active layers across each forward pass during generation, but providing a clear and explicit definition would improve clarity.
* “We allow skips of at most two consecutive layers…” What motivates the need for this specific limitation?
* What does the "Original" column in Table 2 represent?

---

> ### Author Response · Authors · 2025-11-24
>
> We sincerely thank the reviewer for the constructive and detailed feedback. Your comments helped us clarify key technical aspects of Dr.LLM, including the scalability and cost of MCTS supervision (24 h on one A100 GPU), the KV-cache consistency mechanism, and the practical runtime gains shown through wall-clock evaluation (15.3 % end-to-end speedup). These suggestions greatly improved the completeness and clarity of our analysis and strengthened the presentation of Dr.LLM’s efficiency and applicability.
>
> **W1: MCTS cost and data scope**
>
> We appreciate this important point. The offline MCTS supervision is lightweight compared to prior adaptive-depth or fine-tuning approaches. Generating the full training dataset of 4k optimized paths required only 24 hours on a single NVIDIA A100 GPU (8 GB VRAM), equivalent to 961k forward passes, with no backpropagation or optimizer states, as the search uses inference only. In contrast, comparable baselines such as FlexiDepth and Mixture-of-Recursions involve hundreds of thousands of fine-tuning steps on 300k-326k examples, which requires multiple GPUs and several days of training. Router training in Dr.LLM itself is even cheaper, 4 hours on a single A100 (11 M parameters, 0.14 % of model weights).
>
> **W2: KV-cache compatibility**
>
> We appreciate the reviewer’s question. The router does not re-select a path at every decoding step. Instead, the routing decisions are made once at the start of generation and remain fixed for the entire sequence, following the execution paths derived from our MCTS supervision. This ensures that each layer’s activation pattern, and thus its KV cache structure, remains consistent across all tokens. We have clarified this explicitly in line 160 of the revision.
>
> **W3: Router overhead and efficiency**
>
> We thank the reviewer for this suggestion. We have conducted wall-clock measurements to quantify the actual runtime improvement. On sequences of 1,000 generated tokens, Dr.LLM achieves a 15.3% reduction in total generation time (29.21 s vs. 34.49 s for the baseline), while the router adds only 0.27 s of overhead per query (<1% of total latency). This confirms that the lightweight routers introduce negligible cost compared to transformer layers, and the observed layer savings translate directly into real-world speedups. We have incorporated these wall-clock results in the new revision (Introduction, line 60; Experiments, line 302) to explicitly demonstrate that the reported layer savings correspond to real-world latency improvements.
>
> **Q1: Definition of “number of layers used” metric**
>
> You’re absolutely right about the definition. We explicitly clarified in line 58 in the new revision.
>
> **Q2: Constraint on consecutive skips**
>
> Thanks for your comment. We want to clarify that Dr.LLMs can currently skip more than 2 layers (MCTS can select the skip decision several times). We meant that adding a decision "skip the 3 next layers" would waste search time because this decision would rarely be selected (since we found it degrades the accuracy too much).
>
> **Q3: “Original” column in Table 2**
>
> We sample our training data from existing ones (ARC, Dart), so “Original” refers to the size of these existing datasets. We provide these numbers to show that using only a small sample of the data (17% of ARC) is sufficient for significant gains compared to other methods.

---

> > ### Comment · Reviewer_kHKr · 2025-11-24
> >
> > Thank you to the authors for their detailed response.
> >
> > I have one remaining question regarding the router.
> > Is the choice to apply the router only once during inference driven by KV-cache compatibility, or is there another reason?
> >
> > Additionally, do the authors plan to extend the router’s use to later stages of inference in future work? It seems that relying solely on information from the prefill tokens may be somewhat limiting.

---

> ### Author Response · Authors · 2025-11-24
>
> Thanks for your prompt response and interesting future directions.
>
> > Is the choice to apply the router only once during inference driven by KV-cache compatibility, or is there another reason?
>
> The router is applied once per sequence primarily for efficiency and KV-cache compatibility. Making routing decisions once avoids per-token router execution, which would add synchronization and routing overhead at every decoding step. In our current setup, routers contribute less than 1% of total latency (0.27 s overhead on 1k-token generation), and keeping the path fixed ensures that the KV-cache remains consistent across all tokens. Recomputing decisions dynamically would require reinitializing cache entries for multiple layers at every step, negating the measured 15.3 % wall-clock speedup.
>
> > do the authors plan to extend the router’s use to later stages of inference in future work? It seems that relying solely on information from the prefill tokens may be somewhat limiting.
>
> Yes, we plan to extend routing to later decoding stages in future work. We are exploring periodic or adaptive routing, where decisions can be updated after several hundred tokens while maintaining cache consistency and low computational overhead.

---

### Official Review · Reviewer_nSUU · 2025-11-01

**Soundness:** 3
**Presentation:** 3
**Contribution:** 3
**Rating:** 6
**Confidence:** 3

**Summary:**

This paper introduces Dr.LLM, a new method to make Large Language Models (LLMs) more efficient and sometimes more accurate. It works by adding small "routers" to an existing LLM that dynamically decide whether to use, skip, or repeat each layer for a given problem.

**Strengths:**

1. The framework is retrofittable, meaning it can be applied to existing, pre-trained models without costly full-scale retraining. Keeping the base LLM weights frozen makes the approach highly practical and accessible.

2. Using MCTS to generate an "oracle" dataset of optimal execution paths is a very clever way to supervise the routers. This avoids complex reinforcement learning and eliminates the need for a costly search process during actual inference.

3. The claim that the trained routers generalize well to out-of-domain tasks suggests they are learning a robust, transferable policy for allocating computational resources rather than just overfitting to the training data.

**Weaknesses:**

1. The paper emphasizes that router training is lightweight, but the initial offline MCTS process to find the optimal paths could be extremely computationally expensive. The true cost of preparing the training data is not fully addressed.

2. The search space for MCTS (all possible combinations of skipping/repeating layers) grows exponentially with model depth. It's unclear how well this search process scales to extremely deep models (e.g., 100+ layers).

3. The actions (skip, execute, repeat) are discrete and simple. This might not be nuanced enough for problems where only a part of a layer (e.g., a few attention heads) is needed, potentially leaving further efficiency gains on the table.

**Questions:**

Same as above.

---

> ### Author Response · Authors · 2025-11-24
>
> We sincerely thank the reviewer for the insightful feedback. Your comments helped us clarify the low computational cost of MCTS (24 h on a single A100, 8 GB VRAM), emphasize its scalability, and discuss how Dr.LLM complements finer-grained routing approaches, improving both clarity and completeness of the paper.
>
> **W1: Cost of MCTS supervision**
>
> Thanks for your important question. (i) The cost of our MCTS dataset collection is low compared to existing techniques that require training the LLM from scratch or heavy fine-tuning (e.g. FlexiDepth or MoD). From that prospective, Dr.LLM is much cheaper since our search relies only on inference which is highly efficient (KV-cache optimization, no backpropagation, and single copy of the weights in memory unlike the  3 copies of the weights needed by the Adam optimizer) ; (ii) we actually managed to generate the data of all in a single A100 GPU using only 8GBs VRAM in 24 hours, which is tiny compared to pretraining the same model from scratch on millions of data points or finetuning on 300k samples, both time- and memory-wise.
>
> **W2: Scalability of MCTS search**
>
> We appreciate the reviewer’s insightful question. While the theoretical search space of all skip/repeat combinations grows exponentially with model depth, Dr.LLM’s search process remains tractable **because it uses a fixed simulation budget and a length-aware UCB heuristic that prioritizes promising paths early**. The goal is not to exhaustively explore all configurations, but to identify better-than-default execution paths that improve accuracy under a compute budget. In practice, our search covers only a small fraction of possible actions (50 actions out of  3^32 possible actions for LLaMA-8B [32 layers]), as shown in Section 4.2. Yet, it yields consistent improvements, up to +2.5% accuracy with 8.0 layers saved across 3B-8B models. Moreover, the search can be further optimized via weighted sampling, e.g., down-weighting early-layer skips that are rarely beneficial (as observed in Fig. 4). These mechanisms ensure that MCTS scales effectively to deeper models while maintaining efficiency and search quality.
>
> **W3: Token-level actions**
>
> This is a valid point, we focus on sequence-level routing. Token/head-level granularity (e.g., MoD) is complementary and orthogonal. Combining fine-grained (e.g., token/head) and coarse-grained (layer) routing is a great future direction.

---

### Official Review · Reviewer_wfaJ · 2025-11-01

**Soundness:** 3
**Presentation:** 3
**Contribution:** 3
**Rating:** 4
**Confidence:** 3

**Summary:**

This paper proposes Dr.LLM, a retrofittable framework that introduces per-layer dynamic routers for frozen pretrained LLMs. Each router decides whether to skip, execute, or repeat a transformer block, allowing adaptive compute allocation per input. The routers are trained with explicit supervision derived from offline Monte Carlo Tree Search (MCTS), which identifies optimal layer configurations under compute constraints. The approach achieves accuracy improvements on reasoning-heavy tasks (ARC, DART) while reducing the number of executed layers and generalizes well to out-of-domain benchmarks.

**Strengths:**

1. Novel supervised routing framework that avoids inference-time search and large-scale retraining, yet improves both efficiency and accuracy.
2. Strong empirical results showing consistent gains on logic and math benchmarks, plus solid cross-domain generalization with minimal accuracy drop.

**Weaknesses:**

1. Lack of wall-clock evaluation – The reported “layers saved per query” metric does not directly reflect real-world speedup. Since each layer involves router computation overhead, the paper should include wall-time results to substantiate efficiency claims.
2. Limited training exploration – The routers are trained separately from the LLM. Joint training (LLM + routers) could yield better synergy and higher final performance, especially since inference is a long-term process while training is a one-time cost.
3. Unclear batch inference feasibility – It’s uncertain whether Dr.LLM supports efficient batch processing, as per-token dynamic routing can lead to divergent compute paths and reduced parallelism. Discussion or experiments on batch inference scalability would strengthen the work.

**Questions:**

If all concerns shown in weaknesses are resolved, I would raise my score.

---

> ### Author Response · Authors · 2025-11-24
>
> We sincerely thank the reviewer for their thoughtful and constructive feedback. Your comments have directly strengthened the paper by motivating us to add wall-clock latency results (15.3 % end-to-end speedup), clarify batch inference scalability and KV-cache compatibility, and analyze the limitations of joint training. These revisions have improved both the clarity and completeness of Dr.LLM’s empirical validation and practical applicability.
>
> **W1: Latency measurements**
>
> We have conducted wall-clock measurements to quantify the actual runtime improvement. On sequences of 1,000 generated tokens, Dr.LLM achieves a 15.3% reduction in total generation time (29.21 s vs. 34.49 s for the baseline), while the router adds only 0.27 s of overhead per query (<1% of total latency). This confirms that the lightweight routers introduce negligible cost compared to transformer layers, and the observed layer savings translate directly into real-world speedups. We have incorporated these wall-clock results in the new revision (Introduction, line 60; Experiments, line 302) to explicitly demonstrate that the reported layer savings correspond to real-world latency improvements.
>
> **W2: Joint LLM+Routers training**
>
> We appreciate the reviewer’s point regarding joint training. In preliminary experiments, jointly training the LLM and routers led to an 11.8% drop in overall accuracy. Specifically, router validation accuracy decreased from 95.2% to 20.1%, as the router labels, derived from the frozen LLM’s hidden states, became misaligned once the backbone weights were updated. This instability degraded routing quality and overall performance. While large-scale co-training or fine-tuning could help recover synergy (as shown in MoD, FlexiDepth, Mixture of recursions), such experiments require substantially more compute than was available to us. Our focus is instead on developing routers that are lightweight, data-efficient, and retrofittable to existing pretrained LLMs without modifying their weights.
>
> **W3: Batch inference**
>
> We thank the reviewer for highlighting this point. Dr.LLM supports efficient batch inference because routing is performed once per sequence, not per token. The router decides to skip, execute, or repeat each layer based on the hidden states of the input sequence. Consequently, all tokens within a sequence share the same path, and batch elements maintain uniform tensor shapes within each operation, avoiding the divergence and synchronization issues typical of per-token routing.
> In practice, sequences that share the same routing decision at a given layer are processed together as microbatches, ensuring high GPU utilization and static kernel shapes. Because repeat actions are infrequent (2.3% of layers), the synchronization overhead is minimal. This design makes Dr.LLM compatible with standard batched decoding and KV caching, achieving both dynamic depth allocation and parallel efficiency. We will add more results in a new revision in 4 days, clarify these points in the revised manuscript and include a brief discussion of batch inference scalability.

---

> > ### Comment · Reviewer_wfaJ · 2025-11-26
> > **Partially clarify my concerns**
> >
> > Thanks authors for the feedback. I am still concerned with W3. Even though the routing is performed per sequence, in batch inference, different sequences can lead to different paths. Moreover, after prefilling, models generate outputs token-by-token.

---

> > > ### Author Response · Authors · 2025-11-29
> > >
> > > We have now conducted comprehensive experiments that demonstrate Dr.LLM's efficient batch processing capability. We show that Dr.LLM supports batch inference and that Dr.LLM always increases throughput for all batch sizes from 1 to 128.
> > >
> > > **Implementation.**
> > > We perform synchronization per-layer, grouping sequences by their routing decision (skip/execute/repeat) at each layer. At each layer, sequences with the same decision are processed together as a microbatch. When a repeat decision occurs, all sequences must wait for that group to complete both passes through the layer before proceeding. However, the synchronisation overhead is low because repeat actions occur in only 2.3% of layers. The high routing consensus (80-89% agreement) ensures that most layers process the majority of sequences in a single unified batch, maintaining high GPU utilization.
> > >
> > > **Results.**
> > > The table below reports throughput measurements (tokens/second) on ARC-E, ARC-C, and DART 1-5 datasets using LLaMA-3B-Instruct. Dr.LLM achieves an average improvement of 8.65% across all batch sizes. Importantly, Dr.LLM always increases throughput, showing that synchronisation overhead is not a problem even for large batch sizes.
> > >
> > > | Batch Size | Original Throughput | Dr.LLM Throughput | Improvement |
> > > |------------|---------------------|-------------------|-------------|
> > > | 1          | 25.05              | 31.78             | 26.89%      |
> > > | 2          | 37.97              | 41.87             | 10.29%      |
> > > | 4          | 70.74              | 76.12             | 7.60%       |
> > > | 8          | 107.74             | 115.56            | 7.25%       |
> > > | 16         | 117.86             | 122.89            | 4.26%       |
> > > | 32         | 125.43             | 130.83            | 4.31%       |
> > > | 64         | 103.18             | 108.05            | 4.72%       |
> > > | 128        | 109.12             | 113.38            | 3.90%       |
> > > | **Average** |                   |                   | **8.65%**   |
> > >
> > > **Decision Consensus**
> > > The key insight explaining why synchronization overhead remains minimal is that sequences naturally exhibit high routing consensus. We measured inter-prediction similarity across all six models in our study, finding that routing decisions agree 80-89% of the time across different sequences. This high similarity means that at most layers, sequences make identical skip/execute/repeat decisions, allowing our microbatching strategy to group them efficiently.
> > >
> > > | Model              | Inter-Prediction Similarity (%) |
> > > |--------------------|---------------------------------|
> > > | LLaMA-3B-Instruct  | 80.47                          |
> > > | LLaMA-8B-Instruct  | 86.20                          |
> > > | LLaMA-3B-Base      | 80.86                          |
> > > | LLaMA-8B-Base      | 83.49                          |
> > > | Qwen-3B-Instruct   | 89.41                          |
> > > | Qwen-8B-Instruct   | 85.26                          |
> > >
> > > This consensus directly drives batching efficiency. Batches with 95% consensus achieve 100.28 tokens/sec, while perfect consensus reaches 121.58 tokens/sec, demonstrating that higher routing agreement naturally improves parallel processing.
> > >
> > > | Inter-Prediction Similarity (%) | Throughput |
> > > |---------------------------------|------------|
> > > | 75                              | 82.05      |
> > > | 80                              | 85.56      |
> > > | 85                              | 91.43      |
> > > | 90                              | 95.81      |
> > > | 95                              | 100.28     |
> > > | 100                             | 121.58     |
> > >
> > > The practical implication is that despite the per-layer synchronization requirement, the combination of high routing consensus and infrequent repeat decisions allows Dr.LLM to achieve both dynamic depth allocation and parallel efficiency simultaneously. The empirical results demonstrate that the computational savings from skipped and optimally routed layers outweigh any synchronization overhead, resulting in consistent throughput improvements across all batch sizes.

---

### Official Review · Reviewer_p46J · 2025-11-02

**Soundness:** 3
**Presentation:** 3
**Contribution:** 2
**Rating:** 4
**Confidence:** 4

**Summary:**

The paper proposes a method for dynamically skipping, executing or repeating layers in transformer architecture per example in order to improve the model quality while preventing wasted compute on simpler queries. It can be applied to any LLM that is already trained without altering the weights, only adding per-layer routers. The routers are trained using an offline tree-based search method and use hidden layers of the previous layers as inputs. The search algorithm is monte carlo based and includes an explicit length penalty to incentivize shorter paths to favor executing less number of layers. The method is evaluated on Llama and Qwen models and has shown to improve accuracy around 3% while saving 5 layers per example on average.

**Strengths:**

- The method is applicable to already trained LLMs with only minimal router training
- It is a comprehensive method which combines layer skipping and looping which was previously explored separately in most former work
- Research artifacts including the code and data is shared publicly

**Weaknesses:**

- There is no latency measurements shown in the paper.  While the method reduces 5 layers per example on average, the routers add additional overhead per layer and overall latency might increase because of that.
- The method is not applicable to batching as the routing is done per example level, when the batch size is greater than 1, accuracy might suffer.
- Out of domain generalization looks weak. While the method improves accuracy for in-domain tasks (the tasks the routers are trained for), for out of domain tasks accuracy drops.
- I have several questions about the evaluation, listed in questions section below.

**Questions:**

- Figure 1 shows the network has 600+ layers. -> why there are so many layers? No state of the art LLM has that many layers.

- The router accuracy during training is 61%, which seems low. What’s the implication of this, have you tried improving the router accuracy?

- “all four models gain 0.40%p accuracy on GSM8k” -> This claim on page 7 does not match the results on Table 4. Llama 3B accuracy drops on GSM8k according to table 4.

- Why are there no baseline llama numbers in Table 5? It’d be good to add the baseline in this table as some of the tasks in this table were not shown before in the paper. The table shows Dr.LLM is better than the state of the art methods however it does not show how it’s better than the pure LLM in HumanEval, Hellaswag, MMLU tasks.

---

> ### Author Response · Authors · 2025-11-24
>
> We sincerely thank Reviewer for the detailed and constructive feedback. Your comments have substantially improved the clarity and rigor of our work. In response, we have added wall-clock latency measurements (showing a 15.3 % speedup), clarified the batching and KV-cache mechanism, detailed router accuracy (96.8 % per-layer), and expanded our discussion of out-of-domain generalization and experimental consistency. These revisions have made the presentation of Dr.LLM more complete and its practical contributions clearer.
>
> **W1: Latency measurements and routers overhead**
>
> We sincerely thank the reviewer for their insightful feedback. We have conducted wall-clock measurements to quantify the actual runtime improvement. On sequences of 1,000 generated tokens, Dr.LLM achieves a 15.3% reduction in total generation time (29.21 s vs. 34.49 s for the baseline), while the router adds only 0.27 s of overhead per query (<1% of total latency). This confirms that the lightweight routers introduce negligible cost compared to transformer layers, and the observed layer savings translate directly into real-world speedups. We have incorporated these wall-clock results in the new revision (Introduction, line 60; Experiments, line 302) to explicitly demonstrate that the reported layer savings correspond to real-world latency improvements.
>
> **W2: Batch inference**
>
> We thank the reviewer for highlighting this point. Dr.LLM supports efficient batch inference because routing is performed once per sequence, not per token. The router decides to skip, execute, or repeat each layer based on the hidden states of the input sequence. Consequently, all tokens within a sequence share the same path, and batch elements maintain uniform tensor shapes within each operation, avoiding the divergence and synchronization issues typical of per-token routing.
> In practice, sequences that share the same routing decision at a given layer are processed together as microbatches, ensuring high GPU utilization and static kernel shapes. Because repeat actions are infrequent (2.3% of layers), the synchronization overhead is minimal. This design makes Dr.LLM compatible with standard batched decoding and KV caching, achieving both dynamic depth allocation and parallel efficiency. We will add more results in a new revision in 4 days, clarify these points in the revised manuscript and include a brief discussion of batch inference scalability.
>
>
> **W3: Out-of-domain generalization**
>
> Thank you for this valuable observation. While routers are trained only on maths and logic tasks (ARC, DART), they generalize well to eight out-of-domain benchmarks, showing only a -0.85% average accuracy drop (Table 4) while still saving 5 layers per example. This small degradation compares favorably to prior adaptive-depth methods, which typically lose 3-8 %p when transferred out of domain (e.g., FlexiDepth -6.1 %p on GSM8k, MindSkip -7.8 %p).
> The limitation arises from the narrow training distribution, not from the routing mechanism itself. We expect that expanding the MCTS supervision to broader domains (e.g., factual, commonsense, and coding data) will further strengthen transferability. We plan to extend the training corpus in future work to enhance cross-domain robustness.
>
> **Q1: Number of saved layers definition**
>
> Thanks for pointing this out. The “number of layers” in Figure 1 is the sum of layers used for all tokens in an example. Models like LLaMA-3B have 32 layers, when generating long sequences of tokens, the total number of layers used is much higher. We clarified in line 58 in the new revision.
>
> **Q2: Router training accuracy**
>
> The reported 61% corresponds to the macro F1-score, which balances performance across the highly imbalanced classes (skip, execute, repeat). In practice, only about 10% of layers involve skip or repeat actions (≈7.6% skip, 2.3% repeat), making the F1-score conservative relative to the router’s overall accuracy. The actual per-layer accuracy is 96.8%, showing that routers predict correctly in nearly all cases. Most misclassifications are conservative, predicting execute instead of skip or repeat, which merely increases computation slightly but does not reduce task accuracy. Consequently, Dr.LLM maintains consistent accuracy gains while preserving efficiency. We have clarified this point in the revised manuscript (line 293).
>
> **Q3: Clarification of GSM8k average accuracy gain statement**
>
> Our apologies for this oversight: we meant the average gain of 0.40%p across models. We fixed this sentence on line 373 on the new revision.
>
> **Q4: Baseline numbers of LLaMA in table 5**
>
> Thank you for the suggestion. We will update the table in the new revision in 3 days to add Llama and the requested benchmarks.

---

> > ### Comment · Reviewer_p46J · 2025-11-25
> >
> > Thank you for the detailed response.
> >
> > For W2, I meant to ask about multiple sequences to share the same path during the decode phase, what's the synchronization mechanism you use for the ski/repeat actions. As the batch size increase, would not this erase the latency savings. Curious to see your results on this.
> >
> > For Q1, thanks for the explanation, figure makes sense now. Minor comment but it might be more intuitive to show avg number of layers used per token.
> >
> > One additional question, are you planning to open source your code?

---

> > > ### Author Response · Authors · 2025-11-29
> > >
> > > > What's the synchronization mechanism you use for the ski/repeat actions. As the batch size increase, would not this erase the latency savings. Curious to see your results on this.
> > >
> > > We have now conducted comprehensive experiments that demonstrate Dr.LLM's efficient batch processing capability. We show that Dr.LLM supports batch inference and that Dr.LLM always increases throughput for all batch sizes from 1 to 128.
> > >
> > > **Implementation.**
> > > We perform synchronization per-layer, grouping sequences by their routing decision (skip/execute/repeat) at each layer. At each layer, sequences with the same decision are processed together as a microbatch. When a repeat decision occurs, all sequences must wait for that group to complete both passes through the layer before proceeding. However, the synchronisation overhead is low because repeat actions occur in only 2.3% of layers. The high routing consensus (80-89% agreement) ensures that most layers process the majority of sequences in a single unified batch, maintaining high GPU utilization.
> > >
> > > **Results.**
> > > The table below reports throughput measurements (tokens/second) on ARC-E, ARC-C, and DART 1-5 datasets using LLaMA-3B-Instruct. Dr.LLM achieves an average improvement of 8.65% across all batch sizes. Importantly, Dr.LLM always increases throughput, showing that synchronisation overhead is not a problem even for large batch sizes.
> > >
> > > | Batch Size | Original Throughput | Dr.LLM Throughput | Improvement |
> > > |------------|---------------------|-------------------|-------------|
> > > | 1          | 25.05              | 31.78             | 26.89%      |
> > > | 2          | 37.97              | 41.87             | 10.29%      |
> > > | 4          | 70.74              | 76.12             | 7.60%       |
> > > | 8          | 107.74             | 115.56            | 7.25%       |
> > > | 16         | 117.86             | 122.89            | 4.26%       |
> > > | 32         | 125.43             | 130.83            | 4.31%       |
> > > | 64         | 103.18             | 108.05            | 4.72%       |
> > > | 128        | 109.12             | 113.38            | 3.90%       |
> > > | **Average** |                   |                   | **8.65%**   |
> > >
> > > **Decision Consensus**
> > > The key insight explaining why synchronization overhead remains minimal is that sequences naturally exhibit high routing consensus. We measured inter-prediction similarity across all six models in our study, finding that routing decisions agree 80-89% of the time across different sequences. This high similarity means that at most layers, sequences make identical skip/execute/repeat decisions, allowing our microbatching strategy to group them efficiently.
> > >
> > > | Model              | Inter-Prediction Similarity (%) |
> > > |--------------------|---------------------------------|
> > > | LLaMA-3B-Instruct  | 80.47                          |
> > > | LLaMA-8B-Instruct  | 86.20                          |
> > > | LLaMA-3B-Base      | 80.86                          |
> > > | LLaMA-8B-Base      | 83.49                          |
> > > | Qwen-3B-Instruct   | 89.41                          |
> > > | Qwen-8B-Instruct   | 85.26                          |
> > >
> > > This consensus directly drives batching efficiency. Batches with 95% consensus achieve 100.28 tokens/sec, while perfect consensus reaches 121.58 tokens/sec, demonstrating that higher routing agreement naturally improves parallel processing.
> > >
> > > | Inter-Prediction Similarity (%) | Throughput |
> > > |---------------------------------|------------|
> > > | 75                              | 82.05      |
> > > | 80                              | 85.56      |
> > > | 85                              | 91.43      |
> > > | 90                              | 95.81      |
> > > | 95                              | 100.28     |
> > > | 100                             | 121.58     |
> > >
> > > The practical implication is that despite the per-layer synchronization requirement, the combination of high routing consensus and infrequent repeat decisions allows Dr.LLM to achieve both dynamic depth allocation and parallel efficiency simultaneously. The empirical results demonstrate that the computational savings from skipped and optimally routed layers outweigh any synchronization overhead, resulting in consistent throughput improvements across all batch sizes.
> > >
> > > > One additional question, are you planning to open source your code?
> > >
> > > Yes, we plan to release the code and weights upon acceptance.

---

### Author Response · Authors · 2025-12-03
**Final Author Comment to the Area Chair**

**1. Summary of contributions**

Dr.LLM is a retrofittable framework for large language models that:
- Adds small per-layer routers to _frozen_ pretrained LLMs, without changing their weights.
- Lets each router decide to **skip, execute, or repeat** a layer, adapting compute to the input.
- Trains these routers by supervising them with offline MCTS that finds good layer paths under a compute budget, so there is no search at inference time.
- Improves accuracy on reasoning-heavy tasks while reducing the number of executed layers, with good transfer to out-of-domain benchmarks.

**2. Summary of initial reviews**

- **Reviewer p46J (score 4)** and **Reviewer wfaJ (score 4)**:
Found the method sound and clearly presented, and liked that it is retrofittable and improves both efficiency and accuracy. Their main concerns were the lack of (1) wall-clock latency results, (2) batch inference feasibility, and (3) out-of-domain behavior.

- **Reviewer nSUU (score 6)** and **Reviewer kHKr (score 6)**:
Appreciated the clarity, the use of MCTS for routing supervision, and the strong in-domain results. Their main concerns were the (1) cost and scalability of MCTS, (2) KV-cache compatibility, and (3) whether the router overhead justifies the claimed efficiency gains.

Reviewer wfaJ explicitly wrote that they would **raise their score if the weaknesses were resolved**.

**3. Reviewer-specific summaries and discussions**

--> **Reviewer p46J (score 4)**

**Main strengths**
- Applicable to already trained LLMs with only router training.
- Combines skipping and repeating layers in a unified framework.
- Code and data are shared.

**Main concerns**
- No wall-clock latency; routers might negate layer savings.
- Unclear suitability for batch inference.
- Out-of-domain generalization appears weak.
- Clarifications on:
  - Very large “600+ layers” in Figure 1.
  - Low reported router accuracy (61%).
  - GSM8k statement vs Table 4.
  - Missing baseline LLaMA numbers in Table 5.

**Our responses and new results**
- **Latency (W1):**
 We ran wall-clock experiments on sequences of 1,000 generated tokens. Dr.LLM achieves a **15.3% end-to-end speedup** (29.21 s vs 34.49 s), while the router adds **only 0.27 s per query** (<1% of total latency). These results are now reported in the Introduction and Experiments sections.
- **Batch inference (W2 + discussion follow-up):**
We clarified that routing is done per sequence, not per token. All tokens in a sequence share the same path, and we group sequences with the same decision (skip/execute/repeat) into _microbatches_ at each layer.
New experiments on ARC-E, ARC-C, and DART 1-5 (LLaMA-3B-Instruct) show that Dr.LLM always increases throughput across batch sizes 1-128, with an **average improvement of 8.65%**. High routing consensus (80-89% agreement across sequences) keeps synchronization overhead small.

- **Out-of-domain generalization (W3):**
 We measured that routers trained only on math/logic tasks lead to only -0.85% average accuracy drop across eight out-of-domain benchmarks, while still saving 5 layers per example. Dr.LLM improves upon prior methods (e.g., FlexiDepth and MindSkip, which can lose 3-8 percentage points). We discussed that this limitation comes from the narrow training distribution, and that expanding supervision to broader domains is a natural extension.

- **Clarifications to questions (Q1–Q4):**
  - **Figure 1 / “600+ layers”:** We clarified that this counts the _total number of layers used across all tokens in a sequence_, not the physical depth of the model.
  - **Router accuracy:** We explained that 61% is a _macro F1_ across imbalanced classes (skip/execute/repeat), while the true _per-layer accuracy is 96.8%_. Misclassifications are mostly conservative (predicting execute instead of skip/repeat), which may slightly increase compute but does not reduce accuracy.
  - **GSM8k statement:** We corrected the text to state the _average gain of 0.40 percentage points_; the earlier phrasing was misleading.

Reviewer p46J thanked us for the detailed responses and asked about batch synchronization and code release. We provided the throughput table and confirmed that we will open-source code and weights upon acceptance. _Additionally, The discussion with Reviewer p46J was ongoing and constructive; however, the discussion period froze shortly after our final clarifications were posted, and the reviewer did not have the opportunity to update their score or respond further despite expressing openness to raising it._

---

> ### Author Response · Authors · 2025-12-03
>
> --> **Reviewer wfaJ (score 4)**
>
> **Main strengths they highlighted**
> - Novel supervised routing framework using MCTS that avoids inference-time search and large-scale retraining.
> - Strong empirical gains on reasoning tasks and good cross-domain generalization.
>
> **Main concerns**
> - No wall-clock latency evaluation.
> - Limited exploration of joint training of LLM and routers.
> - Batch inference feasibility and impact on parallelism.
>
> They wrote: _“If all concerns shown in weaknesses are resolved, I would raise my score.”_
>
> **Our responses and new results**
> - **Latency (W1):**
> Same experiments as above: **15.3% end-to-end speedup** with negligible (<1%) router overhead. These results directly address the request for real-time speed measurements.
>
> - **Joint training (W2):**
> We ran preliminary joint training of LLM + routers. This led to an 11.8% drop in overall accuracy, with router validation accuracy falling from _95.2% to 20.1%_, because the router labels (computed from the frozen model) became misaligned when the backbone changed.
>  We explain that while large-scale co-training could recover synergy (as in MoD, FlexiDepth, etc.), it would require significantly more compute and complexity, which goes against our goal of a lightweight, retrofittable method that works on fixed pretrained models.
>
> - **Batch inference (W3 + discussion follow-up):**
>  We clarified that routing is **per sequence**, and we introduced **per-layer synchronization with microbatching**, as in the response to Reviewer p46J.
>
> _The discussion with Reviewer wfaJ did not continue after our detailed replies; the discussion period froze before the reviewer could acknowledge our responses or adjust their score, despite explicitly stating they would raise the score if the concerns were resolved._
>
> --> **Reviewer nSUU (score 6)**
>
> **Main strengths they highlighted**
> - Practical retrofit framework for frozen models.
> - Clever use of MCTS to generate oracle paths, avoiding RL and inference-time search.
> - Good out-of-domain generalization, suggesting a robust routing policy.
>
> **Main concerns**
> - The cost of the offline MCTS process used to generate training data may be high.
> - Scalability of MCTS given the large search space.
> - Simple discrete actions (skip/execute/repeat) may miss finer-grained efficiency gains.
>
> **Our responses**
> - **MCTS cost (W1):**
>  We quantified the relatively low cost: generating the full MCTS supervision set took 24 hours on a single A100 GPU (8 GB VRAM), amounting to 961k forward passes (no backpropagation). Router training itself takes about 4 hours on a single A100, with routers comprising only 0.14% of model parameters. We compare this to baselines that require multi-GPU fine-tuning on hundreds of thousands of examples, making Dr.LLM substantially cheaper in practice.
> - **MCTS scalability (W2):**
> We clarified that we do not attempt to exhaustively explore all skip/repeat combinations. We use a fixed simulation budget (e.g., 50 simulations for LLaMA-8B) with a length-aware search strategy that prefers shorter (more efficient) paths. In practice, the search visits only a tiny fraction of the theoretical space but still yields execution paths that give up to +2.5% accuracy with 8 layers saved. We also note that the search can be further optimized by biasing away from actions that rarely help (e.g., skipping many early layers).
> - **Granularity of actions (W3):**
> We positioned Dr.LLM as coarse-grained, layer-level routing that is complementary to finer-grained (token/head) approaches such as MoD. Combining both levels of routing is a promising future direction.
>
> _Reviewer nSUU was already highly positive, and the discussion phase concluded before they had a chance to respond to our detailed answers or further update their strong score._

---

> > ### Author Response · Authors · 2025-12-03
> >
> > --> **Reviewer kHKr (score 6)**
> >
> > **Main strengths they highlighted**
> > - Clear writing and precise presentation.
> > - Unique use of MCTS for routing paths.
> > - Strong performance over prior routing strategies.
> >
> > **Main concerns**
> > - MCTS cost and how it scales if we broaden the router’s training data.
> > - KV-cache compatibility when routing decisions can differ across decoding steps.
> > - Completeness of efficiency analysis given the router overhead.
> > - Clarifying:
> >   - Definition of “number of layers used” per example.
> >   - The rationale for limiting consecutive skips.
> >   - The meaning of the “Original” column in Table 2.
> >
> > **Our responses and follow-up discussion**
> > - **MCTS cost and data scope (W1):**
> > As for Reviewer nSUU, we report that all MCTS supervision was generated in 24 hours on a single A100 (8 GB), and that we train routers on a small subset of available data (e.g., 17% of ARC), yet still obtain significant gains. We explicitly compare this cost to prior work that requires heavy fine-tuning.
> >
> > - **KV-cache compatibility (W2 + follow-up question):**
> >  We clarified that Dr.LLM **computes routing decisions once per sequence**, at the start of generation, and keeps the path fixed for all tokens. This keeps KV-cache usage simple and consistent and avoids any need to recompute caches mid-generation.
> >  In the follow-up discussion, we explained that this design is driven by both efficiency and cache consistency: per-token routing would add router overhead at every step and complicate cache management, likely erasing the measured 15.3% speedup. We also mentioned that we plan to explore periodic or adaptive routing later in the sequence as future work, while maintaining manageable overhead.
> >
> > - **Router overhead and efficiency (W3):**
> >  The wall-clock experiments directly address this: Dr.LLM reduces total generation time by 15.3%, while routers contribute <1% of latency. This shows that the overhead is indeed negligible compared to the savings from skipped layers.
> >
> > - **Clarifications (Q1–Q3):**
> >   - **“Number of layers used”:** We now explicitly define this metric as the sum of active layers across all tokens in a sequence.
> >   - **Consecutive skips:** We clarified that the model can effectively skip more than two layers via repeated skip decisions. We only avoid adding a special “skip 3 layers at once” action because it rarely helps and would waste search budget.
> >   - **“Original” column in Table 2:** We clarified that it refers to the size of the original datasets (ARC, DART) from which we sample training data, showing that a small subset is enough for Dr.LLM to perform well.
> >
> > _Reviewer kHKr asked one clarification question during the discussion, which we fully addressed; however, the reviewer did not have time to acknowledge the follow-up or update the score before the discussion was frozen._
> >
> > **4. Summary of improvements to the paper**
> >
> > Thanks to the reviewers’ feedback and the discussion phase, we have:
> > - Added **wall-clock latency results** demonstrating a **15.3% end-to-end speedup** with negligible router overhead.
> > - Added **batch inference experiments** (throughput vs batch size 1–128) and **routing consensus analysis**, showing consistent throughput gains and explaining why synchronization overhead remains low.
> > - Quantified the **cost of MCTS supervision and router training** and compared it to prior fine-tuning-based methods.
> > - Clarified the **KV-cache compatibility** by stating that routing decisions are fixed per sequence.
> > - Clarified router **accuracy metrics** (macro F1 vs true per-layer accuracy), and how conservative errors affect compute but not accuracy.
> > - Fixed the **GSM8k statement**, added **missing baseline LLaMA numbers**, and clarified key definitions such as “number of layers used” and the “Original” column.
> > - Discussed **joint training** as a future direction and reported initial experiments showing why we focus on frozen backbones.
> > - Outlined **future extensions** (broader training domains, periodic routing during decoding, possible combination with finer-grained routing).
> >
> > These changes make the paper more transparent about efficiency, scalability, and deployment practicality.
> >
> > **5. Concluding remark**
> >
> > In summary, the initial reviews found Dr.LLM to be a clear and promising framework that improves both accuracy and efficiency for frozen LLMs. The main reservations centered on practical deployment issues (latency, batching, cache usage) and the cost of MCTS supervision. Through new experiments and clarifications, we have addressed these concerns one by one and strengthened the empirical and practical case for Dr.LLM as a simple, retrofittable, and deployable approach to dynamic depth in LLMs.

---

### Meta-Review · Area_Chair_B6aX · 2026-01-07

**Summary:**

This paper proposes a method that integrates routing with MCTS to improve efficiency and accuracy in long-context inference. Reviewers generally agree that the method is readily applicable and the empirical results are promising. Several concerns were raised regarding latency, scalability of MCTS, batch inference feasibility. Many of these were addressed in the rebuttal through additional measurements and clarifications (the feedback from Reviewer p46J, wfaJ, and kHKr suggests that most points were resolved). Overall, given the solid motivation and the encouraging empirical performance, I recommend acceptance.

**Reviewer Concerns:**

Reviewer p46J:
- Missing latency measurements; This is explicitly responded with quantitative evaluation. It was not raised again in the followup reviewer feedback, so I suppose this concern was well addressed.
- Incompatible with batch inference;
- Weak OOD generalization;

Reviewer wfaJ:
- Lack of wall-clock evaluation; This is similar to W1 of Reviewer p46J.
- Joint training of the model and routers not exploreed; Author rebuttal provided a explanation.
- Unclear batch inference feasibility; The reviewer was not fully convinced by author rebuttal and questioned the batch inference scalability. After reading the discussion and the paper, I believe the additional clarification from authors largely address the concern.

Reviewer nSUU:
- MCTS process is computational costly & its scalability;

Reviewer kHKr (Reviewer feedback suggests that most of these concerns were addressed.):
- computational cost of MCTS (similar with Reviewer nSUU);
- KV cache compatibility;
- Incomplete efficiency analysis;

**Reviewer Scores:**

Reviewer p46J's followup feedback suggests that most of their concern was addressed.
Reviewer wfaJ indicated that they would raise score if all concerns in weaknesses are resolved. While the discussion ended early, the rebuttal and followup feedback suggest that these concerns were largely addressed.
Given this, my best guess is that reviewers would have tended to move their scores upward had they been able to fully participate in the discussion.

---

### Decision · Program_Chairs · 2026-01-26

Accept (Poster)